# LLM-Guided Evolutionary Program Synthesis for Quasi-Monte Carlo Design

**Amir Sadikov**
University of California, San Francisco
amir.sadikov@ucsf.edu

## Abstract

Low-discrepancy point sets and digital sequences underpin quasi-Monte Carlo (QMC) methods for high-dimensional integration. We cast two long-standing QMC design problems as program synthesis and solve them with an LLM-guided evolutionary loop that mutates and selects code under task-specific fitness: (i) constructing finite 2D/3D point sets with low star discrepancy, and (ii) choosing Sobol' direction numbers that minimize randomized QMC error on downstream integrands. Our two-phase procedure combines constructive code proposals with iterative numerical refinement. On finite sets, we rediscover known optima in small 2D cases and set new best-known 2D benchmarks for $N \geq 40$, while matching most known 3D optima up to the proven frontier ($N \leq 8$) and reporting improved 3D benchmarks beyond. On digital sequences, evolving Sobol' parameters yields consistent reductions in randomized quasi-Monte Carlo (rQMC) mean-squared error for several 32-dimensional option-pricing tasks relative to widely used Joe–Kuo parameters, while preserving extensibility to any sample size and compatibility with standard randomizations. Taken together, the results demonstrate that LLM-driven evolutionary program synthesis can automate the discovery of high-quality QMC constructions, recovering classical designs where they are optimal and improving them where finite-N structure matters. Data and code are available at https://github.com/hockeyguy123/openevolve-star-discrepancy.

## 1 Introduction

Numerical integration in high dimensions is a cornerstone of modern science and engineering. While standard Monte Carlo (MC) methods offer a robust approach, their convergence rate, governed by the Central Limit Theorem, is often insufficient for applications requiring high precision (Glasserman, 2003). Quasi-Monte Carlo (QMC) methods provide a compelling alternative by replacing pseudorandom samples with deterministic, highly uniform point sets (Dick & Pillichshammer, 2010; Owen, 1995). The uniformity of these sets is quantified by their discrepancy, with lower values corresponding to more evenly distributed points. The Koksma–Hlawka inequality provides the theoretical underpinning for QMC, guaranteeing that the integration error is bounded by the product of the variation of the integrand and the star discrepancy of the point set (Koksma, 1964; Hlawka, 1961).

This relationship has fueled decades of research into the discovery and construction of low-discrepancy sets. A primary challenge in this field is the "*ab initio*" construction of a finite point set of $N$ points in $d$ dimensions that minimizes star discrepancy. This is a problem of combinatorial complexity, and while optimal solutions have been found in 2D for $N \leq 21$ and in 3D for $N \leq 8$ (Clément et al., 2024), the problem remains largely open for larger $N$ and higher dimensions.

A related challenge is the construction of infinite low-discrepancy sequences, such as those by Halton, Hammersley, and Sobol'. Among these, Sobol' sequences are particularly prominent due to their excellent distribution properties and efficient generation (Sobol', 1967). However, their quality is highly dependent on a set of integer parameters known as direction numbers, and finding optimal parameters that ensure uniformity across all low-dimensional projections is a difficult combinatorial search problem (Joe & Kuo, 2008).

Recent breakthroughs in Large Language Models (LLMs) have demonstrated their remarkable capabilities in code generation, logical reasoning, and pattern recognition (Gemini Team et al., 2025). This has motivated the development of automated scientific discovery systems, such as AlphaEvolve, which leverage LLMs to navigate complex search spaces (Novikov et al., 2025). In this work, we utilize the OpenEvolve framework (Sharma, 2025), an open-source implementation based on the principles of AlphaEvolve, to tackle the aforementioned challenges in discrepancy theory. We treat the construction of these mathematical objects as a program synthesis problem within an evolutionary framework (Koza, 1994). The LLM acts as an intelligent mutation operator, iteratively modifying code that generates candidate solutions based on feedback from a fitness function.

This paper presents the successful application of this LLM-driven evolutionary approach to two fundamental problems:

1. Discovering finite 2D and 3D point sets with state-of-the-art low star discrepancy.
2. Searching for superior Sobol' direction numbers to minimize randomized quasi-Monte Carlo (rQMC) integration error in the high-dimensional context of pricing exotic financial options (Paskov & Traub, 1995).

Our results show that this methodology can match and, in several important cases, surpass the best known human-derived solutions, without any task-specific training of the LLM. This suggests that LLM-driven evolutionary search is a promising new paradigm for exploration and discovery in computational mathematics.

## 2 THEORETICAL BACKGROUND

### 2.1 STAR DISCREPANCY

Star discrepancy is the most common measure for quantifying the uniformity of a point set within the $d$-dimensional unit hypercube, $[0, 1]^d$ (Owen, 1995). It captures the largest deviation between the volume of an axis-aligned "anchor box" and the fraction of points contained within it.

**Definition 1 (Star Discrepancy).** Let $P = \{\mathbf{x}_1, \ldots, \mathbf{x}_N\}$ be a set of $N$ points in $[0, 1]^d$. An anchor box $[\mathbf{0}, \mathbf{q})$ for any $\mathbf{q} = (q_1, \ldots, q_d) \in [0, 1]^d$ is the hyperrectangle $[0, q_1) \times \cdots \times [0, q_d)$. The star discrepancy $D_N^*$ of the set $P$ is defined as:

$$D_N^*(P) = \sup_{\mathbf{q} \in [0,1]^d} \left| \frac{\#(P \cap [\mathbf{0}, \mathbf{q}))}{N} - \text{Vol}([\mathbf{0}, \mathbf{q})) \right| \tag{1}$$

Here, the supremum is taken over all possible anchor boxes. A small $D_N^*$ value implies that for any anchor box, the fraction of points falling within it is a good approximation of its volume, indicating high uniformity. The practical importance of star discrepancy for QMC is cemented by the Koksma–Hlawka inequality (Koksma, 1964; Hlawka, 1961), which bounds the error of QMC integration:

$$\left| \int_{[0,1]^d} f(\mathbf{u}) \, d\mathbf{u} - \frac{1}{N} \sum_{i=1}^{N} f(\mathbf{x}_i) \right| \leq V(f) \cdot D_N^*(P) \tag{2}$$

where $V(f)$ is the total variation of the function $f$ in the sense of Hardy and Krause (Niederreiter, 1992; Dick & Pillichshammer, 2010). This inequality guarantees that point sets with lower star discrepancy lead to smaller integration error bounds.

### 2.2 SOBOL' SEQUENCES AND DIRECTION NUMBERS

Sobol' sequences are a class of low-discrepancy sequences that are particularly effective for QMC integration (Sobol', 1967; Dick & Pillichshammer, 2010). They are constructed using the properties of primitive polynomials over the finite field of two elements, $\mathbb{F}_2$.

**Definition 2 (Sobol' Sequence Construction).** For each dimension $j \geq 1$, a primitive polynomial over $\mathbb{F}_2$ of degree $s_j$ is chosen (Dick & Pillichshammer, 2010):

$$P_j(z) = z^{s_j} + a_{1,j} z^{s_j - 1} + \cdots + a_{s_j - 1, j} z + 1 \tag{3}$$

where the coefficients $a_{k,j}$ are either 0 or 1. From this polynomial, a sequence of positive, odd integers called direction numbers $m_{k,j}$ (for $k = 1, \ldots, s_j$) are chosen freely.

For $k > s_j$, subsequent direction numbers (expressed as fixed-point binary fractions $v_{k,j} = m_{k,j}/2^k$) are generated by the canonical Sobol' recurrence (Joe & Kuo, 2008; Dick & Pillichshammer, 2010):

$$v_{k,j} \;=\; a_{1,j}v_{k-1,j} \;\oplus\; a_{2,j}v_{k-2,j} \;\oplus\; \cdots \;\oplus\; a_{s_j-1,j}v_{k-s_j+1,j} \;\oplus\; \left(v_{k-s_j,j} \gg s_j\right) \tag{4}$$

where $\oplus$ denotes bitwise XOR on the binary fixed-point representation and $\gg s_j$ is a right bit-shift by $s_j$ places. For $j = 1$, we set $v_{k,1} = 2^{-k}$ (van der Corput base-2).

The $j$-th coordinate of the $i$-th point in the sequence, $x_{i,j}$, is then generated using Gray-code bits:

$$x_{i,j} \;=\; \bigoplus_{k\geq 1} g_k\, v_{k,j}, \qquad \text{with } i = \sum_{k\geq 1} i_k 2^{k-1}, \;\; g \;=\; i \oplus (i \gg 1) \;=\; \sum_{k\geq 1} g_k 2^{k-1}. \tag{5}$$

The quality of the Sobol' sequence, particularly the uniformity of its low-dimensional projections, is critically dependent on the choice of the primitive polynomials and the initial direction numbers $(m_1, \ldots, m_s)$. The work of Joe & Kuo (2008) provides a widely used set of these parameters that serve as a strong baseline. In tables we encode the polynomial coefficients $a_{k,j}$ as an integer $A_j$ (binary bitmask).

## 3 RELATED WORK

Classical approaches for generating low-discrepancy sets are primarily number-theoretic. Foundational methods include Halton, Hammersley, and Sobol' sequences, which are designed to achieve superior asymptotic uniformity compared to random sampling. While powerful, these classical constructions are not always optimal for a finite number of points $N$. Mathematical programming has been used to find provably optimal sets, though these approaches are computationally intensive and limited to small instances (Clément et al., 2024). Heuristic methods, such as genetic algorithms and threshold accepting, have been applied to tackle larger instances by searching the space of point configurations (Clément et al., 2023).

More recently, machine learning techniques have been introduced to this domain. Message-Passing Monte Carlo (MPMC) leverages Graph Neural Networks (GNNs) to transform random initial points into low-discrepancy configurations, achieving state-of-the-art results by directly optimizing point coordinates (Rusch et al., 2024). Our work differs by framing the task as a program synthesis problem rather than direct coordinate optimization.

A related line of research focuses on optimizing the parameters of Sobol' sequences. The quality of a Sobol' sequence is critically dependent on a set of initialization parameters known as direction numbers. The direction numbers published by Joe & Kuo (2008) are a widely-used standard, derived from an extensive computational search to find parameters that ensure high uniformity in two-dimensional projections by minimizing a quality measure known as the $t$-value. Subsequent work has focused on further improving these parameters or guaranteeing quality for specific projection properties crucial for applications such as computer graphics (Bonneel et al., 2025). LatNetBuilder (L'ecuyer & Munger, 2016) generalizes these ideas by searching the parameters of lattice rules and digital nets to minimize user-specified criteria, such as discrepancy bounds or projection-dependent $t$-values.

Our approach is most closely related to the emerging paradigm of using Large Language Models (LLMs) for automated scientific discovery. Novikov et al. (2025) introduces AlphaEvolve, a framework that combines LLMs with an evolutionary search, treating algorithm discovery as a program evolution problem where the LLM functions as an intelligent mutation operator and receives feedback from a fitness function. This method has successfully discovered novel algorithms for fundamental problems, from matrix multiplication (Fawzi et al., 2022) to open mathematical conjectures (Romera-Paredes et al., 2024). Our work is directly inspired by these principles, applying a similar evolutionary loop to the specific mathematical challenges of discovering low-discrepancy sets and optimizing Sobol' sequences.

## 4 METHODOLOGY

Our approach is based on the OpenEvolve framework, an open-source implementation of the principles demonstrated by AlphaEvolve (Novikov et al., 2025). It frames the search for novel mathematical constructs as an evolutionary search over a population of programs that generate them (Koza, 1994). The LLM serves as a sophisticated mutation operator, guided by a fitness function.

The evolutionary loop (Romera-Paredes et al., 2024) proceeds as follows:

1. **Initialization:** The process begins with a population of "parent" programs. These programs are code snippets in Python that generate a candidate solution. The initial population can be seeded with simple heuristics or well-known constructions.

2. **Evaluation:** Each program in the population is executed, and its output is evaluated by a fitness function. The fitness function returns a scalar score quantifying the quality of the solution (e.g., lower rQMC MSE or star discrepancy is better).

3. **Selection and Prompting:** High-performing programs are selected to serve as parents. A detailed prompt is then constructed for the LLM, including the parent program's source code, its fitness score, and an instruction tasking the LLM with generating a variation that will improve upon the score. The prompt also includes code from other high-performing "inspirations" to encourage crossover as well as guidance from the user (Appendix A).

4. **Generation (Mutation):** The LLM receives the prompt and generates a new, modified program. This is the core "mutation" step. The LLM's ability to understand code syntax and semantics allows for complex and intelligent modifications.

5. **Loop:** The newly generated program is evaluated, its fitness is scored, and it is added to the population. The process then repeats, iteratively refining the population toward better solutions.

### 4.1 EXPERIMENTAL SETUP

We used an LLM-guided evolutionary search over a population of Python programs (population 60, four islands with occasional migration). Parents were chosen by fitness with occasional archive sampling; the LLM rewrote core functions each generation. We used Google's Gemini 2.0 Flash as the mutation operator with temperature $0.7$ and top-$p = 0.95$. We did not train or fine-tune the LLM in any way; it is used as an off-the-shelf code rewriter. Each evolutionary run used roughly 2000 LLM calls and a fixed budget for fitness evaluations. Full settings, including population structure, archive, migration, and program validation checks, are given in Appendix A.

### 4.2 DISCOVERY OF LOW-DISCREPANCY POINT SETS

A two-phase strategy was employed to balance broad exploration with fine-tuned optimization.

- **Phase I: Direct Construction:** The LLM was prompted to generate Python code that directly constructs an $N$-point set in a $d$-dimensional space. The initial parent program in 2D implemented a simple shifted Fibonacci lattice (Appendix A, Listing 3) and in 3D implemented a scrambled Sobol' sequence (Appendix A, Listing 4). This phase encouraged the LLM to explore a wide range of constructive heuristics.

- **Phase II: Iterative Optimization:** After a sufficient number of iterations, the LLM was prompted to generate Python code that uses iterative optimization routines (e.g., `scipy.optimize.minimize`) to refine an initial guess. This shifted the search from finding explicit constructions to a direct optimization of the point coordinates.

- **Fitness Function:** The fitness score was $\frac{1}{1+D_N^*}$ where $D_N^*$ is the star discrepancy of the generated point set.

We compute $D_N^*$ exactly by scanning extremal anchor boxes on the coordinate-induced grid with computational complexity $\mathcal{O}(N^{d/2+1})$. Implementation details are in Appendix A.

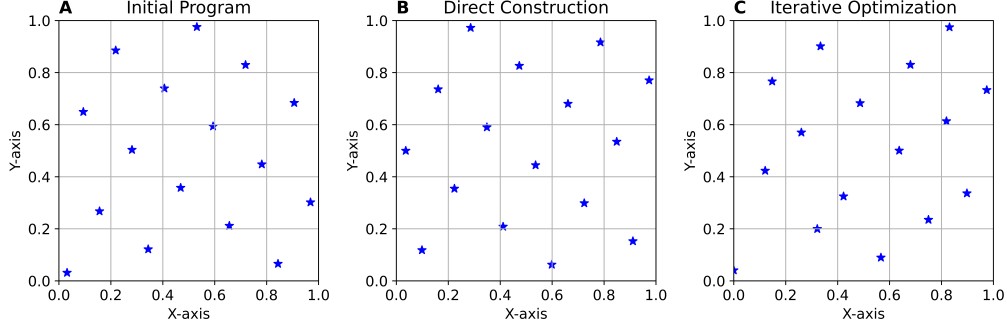

Figure 1: Visualization of $N = 16$ point set generation in two dimensions. (A) Initial shifted Fibonacci lattice (Discrepancy: 0.0962). (B) Best direct construction found in Phase I (Discrepancy: 0.0924). (C) Final optimized point set from Phase II (Discrepancy: 0.0744), which is within 0.68% of the known optimal value of 0.0739.

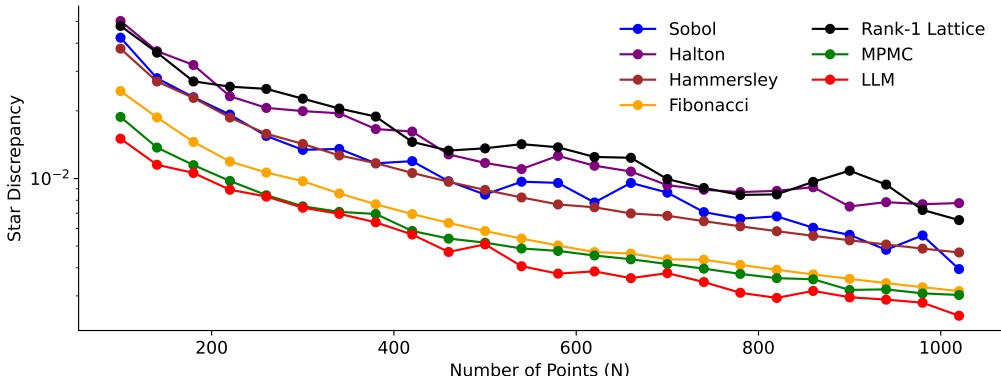

Figure 2: The Star Discrepancy $D_N^*$ of Sobol', Halton, Hammersley, Fibonacci, Rank-1-Lattice, MPMC (message passing Monte Carlo), and LLM-evolved sets for increasing number of points $N = 100 \ldots 1020$ in 2D.

## 4.3 DISCOVERY OF SOBOL' DIRECTION NUMBERS

- **Program Representation:** The evolved programs are Python functions that return a list of dictionaries. Each dictionary contains the Sobol' parameters (`s`, `a`, `m_i`) for a single dimension.

- **Initialization:** The initial population contains the following implementation of the direction numbers (Joe & Kuo, 2008).

- **Fitness Function:** The primary fitness metric is $\frac{1}{1+\text{MSE}}$ where MSE is the mean squared error of an rQMC estimate for a 32-dimensional Asian option price. The MSE is calculated for $N = 8192$ points and averaged over 1000 consistent randomizations (left matrix scramble followed by a Cranley–Patterson random shift, i.e., LMS+shift) to ensure robustness and reproducibility. All rQMC comparisons are paired by using identical randomization seeds per method and $N$. The diffusion paths are constructed from $[0, 1]^d$ via standard time discretization of geometric Brownian motion with equal timesteps; we report this mapping to clarify effective-dimension effects.

## 5 EXPERIMENTS AND RESULTS

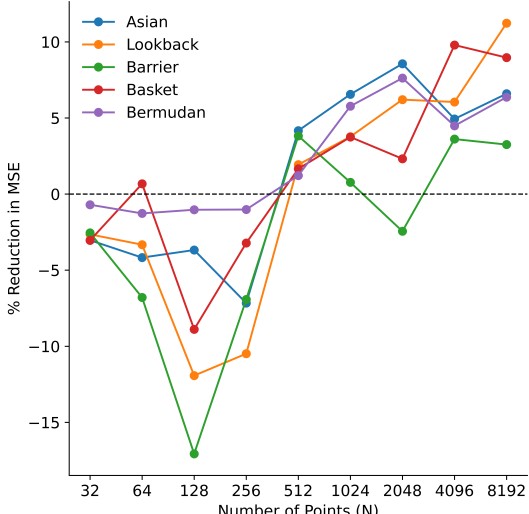

Figure 3: The % reduction in MSE (rQMC integration over 10000 random scrambles and shifts) using Sobol' direction numbers found via LLM evolutionary search vs. those of Joe & Kuo (2008). The % reductions in MSE are averaged across all scenarios of that particular option (Appendix C).

---

**Listing 1** Directly Constructed 16 Point Set ($N = 16$)

```
1  def construct_star():
2      A = np.zeros((N, 2))
3      phi=(math.sqrt(5)-1)/2
4      for i in range(N):
5          A[i, 0]=(i+(1/math.sqrt(3)))/N
6          A[i, 1]=((0.5+(i*phi)%1))%1
7      return A
```

---

## 5.1 DISCOVERY OF LOW-DISCREPANCY POINT SETS

We first applied our two-phase strategy to the canonical problem of discovering $N$-point sets in 2D and 3D with minimal star discrepancy. We illustrate the discovery process for a 2D 16-point set (Fig. 1). The initial program, a Fibonacci lattice, had a discrepancy of 0.0962. After 243 iterations in the direct construction phase, a new construction was found with a discrepancy of 0.0924 (Listing 1), which consisted of fine-tuning optimal shifts to the Fibonacci lattice. After switching to the iterative optimization phase, the framework further refined the point set, achieving a final discrepancy of 0.0744, which is within 0.68% of the known optimal value of 0.0739. The final program creates an initial guess, consisting of a randomly jittered Fibonacci lattice, followed by a sequential least squares programming (SLSQP) optimization loop with stochastic restarts (Listing 2).

We then benchmarked our method against Fibonacci, MPMC (Rusch et al., 2024), and known optimal point sets (Clément et al., 2024) for $N = 1 \ldots 100$ (Table 1). Our method successfully rediscovers the known optimal point sets for $N \leq 10$ and remains highly competitive for larger $N$. The most significant results came from searching for larger point sets where optimal solutions are not known. For 2D point sets with $N > 30$, LLM evolutionary search discovered new configurations with lower star discrepancy than the best-known values from the literature. For instance, for $N = 100$, our method found a point set with a discrepancy of 0.0150, a substantial improvement over the previous best of 0.0188. For $N = 100$ we further compare the full evolutionary loop against a single one-shot prompt and a multi-turn prompting baseline without evolution, as well as against a smaller Gemini Flash-Lite model. Over 16 seeds, evolutionary search with either LLM achieves consistently lower discrepancy with much smaller variance than the prompting-only baselines (Appendix B), showing that population-based evolution is more robust than repeated prompting.

We generated 2D point sets up to $N = 1020$ with lower star discrepancy than previously reported (Appendix B). In 3D, our method matched the known optimal point sets for $N = 1, 2, 3, 5, 6, 7$

| N | Fibonacci | MPMC | LLM | Clément et al. |
|---|---|---|---|---|
| 1 | 1.0000 | **0.6180** | **0.6180** | **0.6180** |
| 2 | 0.6909 | **0.3660** | **0.3660** | **0.3660** |
| 3 | 0.5880 | **0.2847** | **0.2847** | **0.2847** |
| 4 | 0.4910 | **0.2500** | **0.2500** | **0.2500** |
| 5 | 0.3528 | **0.2000** | **0.2000** | **0.2000** |
| 6 | 0.3183 | 0.1692 | **0.1667** | **0.1667** |
| 7 | 0.2728 | 0.1508 | **0.1500** | **0.1500** |
| 8 | 0.2553 | 0.1354 | **0.1328** | **0.1328** |
| 9 | 0.2270 | 0.1240 | **0.1235** | **0.1235** |
| 10 | 0.2042 | 0.1124 | **0.1111** | **0.1111** |
| 11 | 0.1857 | 0.1058 | 0.1039 | **0.1030** |
| 12 | 0.1702 | 0.0975 | 0.0960 | **0.0952** |
| 13 | 0.1571 | 0.0908 | 0.0892 | **0.0889** |
| 14 | 0.1459 | 0.0853 | 0.0844 | **0.0837** |
| 15 | 0.1390 | 0.0794 | 0.0791 | **0.0782** |
| 16 | 0.1486 | 0.0768 | 0.0745 | **0.0739** |
| 17 | 0.1398 | 0.0731 | 0.0712 | **0.0699** |
| 18 | 0.1320 | 0.0699 | 0.0676 | **0.0666** |
| 19 | 0.1251 | 0.0668 | 0.0654 | **0.0634** |
| 20 | 0.1188 | 0.0640 | 0.0611 | **0.0604** |
| 30 | 0.0792 | N/A | 0.0438 | **0.0424** |
| 40 | 0.0638 | N/A | **0.0331** | 0.0332 |
| 50 | 0.0531 | N/A | **0.0278** | 0.0280 |
| 60 | 0.0442 | 0.0273 | **0.0234** | 0.0244 |
| 100 | 0.0275 | 0.0188 | **0.0150** | 0.0193 |

Table 1: 2D Star Discrepancy Comparison for $N = 1 \ldots 100$ between Fibonacci, MPMC (Message-Passing Monte Carlo), LLM evolutionary search, and Clément et al. (provably optimal for $N \leq 20$). Best values are **bolded**.

---

**Listing 2** Iteratively Optimized 2D Point Set ($N = 16$)

```
1   def construct_star():
2       x = np.zeros((N, 2))
3       for i in range(N):
4           x[i, 0] = (i + np.random.rand()) / N
5           x[i, 1] = ((i * 0.38196601125) % 1) + np.random.rand()/(2*N)
6       def discrepancy_wrapper(x):
7           points = x.reshape(N, 2)
8           return star_discrepancy(points)
9       x0 = x.flatten()
10      bounds = [(0.0, 1.0)] * (N * 2)
11      best_result = None
12      best_discrepancy = float('inf')
13      for _ in range(25):
14          x0_restart = x.flatten() + np.random.normal(0, 0.01, N * 2)
15          x0_restart = np.clip(x0_restart, 0.0, 1.0)
16          result = minimize(discrepancy_wrapper, x0_restart, method='
                SLSQP', bounds=bounds, options={'maxiter': 30000, 'ftol':
                1e-15, 'iprint': 0})
17          discrepancy = discrepancy_wrapper(result.x)
18          if discrepancy < best_discrepancy:
19              best_discrepancy = discrepancy
20              best_result = result
21      optimized_points = best_result.x.reshape(N, 2)
22      return optimized_points
```

---

(Table 2) and provided explicit constructions that set new best-known star-discrepancy benchmarks beyond the proven-optimal range for $N > 8$ (Appendix B).

| N | 1 | 2 | 3 | 4 | 5 | 6 | 7 | 8 |
|---|---|---|---|---|---|---|---|---|
| MPMC | 0.6833 | **0.4239** | 0.3491 | 0.3071 | 0.2669 | 0.2371 | 0.2158 | 0.1993 |
| LLM | 0.6823 | **0.4239** | 0.3445 | 0.3042 | **0.2618** | **0.2326** | **0.2090** | 0.1937 |
| Clément et al. | 0.6823 | **0.4239** | 0.3445 | 0.3038 | **0.2618** | **0.2326** | **0.2090** | 0.1875 |

Table 2: 3D Star Discrepancy for $N = 1 \ldots 8$. Lower is better; best per $N$ is **bolded**.

| N | Training Example | | | Out-of-the-Money | | | At-the-Money | | |
|---|---|---|---|---|---|---|---|---|---|
| | Sobol | LLM | p-value | Sobol | LLM | p-value | Sobol | LLM | p-value |
| 32 | **0.248** | 0.252 | 0.976 | **0.192** | 0.198 | 0.976 | **0.325** | 0.335 | 0.976 |
| 64 | **0.0642** | 0.0665 | 0.998 | **0.0605** | 0.0631 | 0.998 | **0.0873** | 0.0908 | 0.998 |
| 128 | **0.0183** | 0.0192 | 1.00 | **0.0220** | 0.0231 | 1.00 | **0.0277** | 0.0282 | 1.00 |
| 256 | **0.00560** | 0.00610 | 1.00 | **0.00860** | 0.00890 | 1.00 | **0.00920** | 0.00980 | 1.00 |
| 512 | 0.00165 | **0.00161** | 0.284 | 0.00347 | **0.00331** | **0.0208** | 0.00325 | **0.00311** | **0.0208** |
| 1024 | 5.42e-04 | **5.24e-04** | 0.0548 | 0.00146 | **0.00131** | **3.5e-08** | 0.00123 | **0.00117** | **0.0304** |
| 2048 | 2.33e-04 | **2.25e-04** | **0.0199** | 6.19e-04 | **5.35e-04** | **1.5e-14** | 5.50e-04 | **5.21e-04** | **0.0131** |
| 4096 | 9.88e-05 | **9.35e-05** | **0.0058** | 2.58e-04 | **2.43e-04** | **4.5e-04** | 2.40e-04 | **2.26e-04** | **0.0019** |
| 8192 | 4.52e-05 | **4.10e-05** | **7.0e-06** | 1.17e-04 | **1.07e-04** | **1.0e-06** | 1.02e-04 | **9.52e-05** | **0.0100** |

| N | In-the-Money | | | High Volatility | | | Low Volatility | | |
|---|---|---|---|---|---|---|---|---|---|
| | Sobol | LLM | p-value | Sobol | LLM | p-value | Sobol | LLM | p-value |
| 32 | **0.140** | 0.143 | 0.976 | **2.08** | 2.15 | 0.976 | **0.0238** | 0.0246 | 0.976 |
| 64 | **0.0367** | 0.0382 | 0.998 | **0.571** | 0.596 | 0.998 | **0.00660** | 0.00680 | 0.998 |
| 128 | **0.0106** | 0.0111 | 1.00 | **0.181** | 0.188 | 1.00 | **0.00217** | 0.00221 | 1.00 |
| 256 | **0.00308** | 0.00337 | 1.00 | **0.0598** | 0.0643 | 1.00 | **7.56e-04** | 8.03e-04 | 1.00 |
| 512 | 8.68e-04 | **8.59e-04** | 0.284 | 0.0205 | **0.0196** | **0.0073** | 2.85e-04 | **2.75e-04** | **0.0208** |
| 1024 | 2.61e-04 | **2.60e-04** | 0.258 | 0.00757 | **0.00706** | **3.4e-04** | 1.13e-04 | **1.09e-04** | 0.0717 |
| 2048 | 1.01e-04 | **1.00e-04** | 0.388 | 0.00315 | **0.00287** | **1.8e-06** | 5.21e-05 | **4.96e-05** | **0.0056** |
| 4096 | 4.04e-05 | **3.95e-05** | 0.218 | 0.00129 | **0.00123** | **0.0022** | 2.32e-05 | **2.20e-05** | **5.7e-04** |
| 8192 | 1.77e-05 | **1.68e-05** | **0.0047** | 5.66e-04 | **5.32e-04** | **0.0156** | 1.01e-05 | **9.30e-06** | **1.3e-04** |

Table 3: Mean Squared Error (MSE) and p-values for Asian option scenarios. The table compares the standard Sobol' sequence against a sequence found via LLM evolutionary search (LLM). P-values are from a one-sided Wilcoxon signed-rank test conducted over the 10000 randomizations with pairing by identical randomization seeds and are false discovery rate (FDR) corrected. P-values below 0.05 are **bolded**.

One natural concern is that Phase II might simply be rediscovering standard coordinate-wise local optimization, with the LLM contributing little beyond calling `scipy.optimize.minimize`. To test this, we ran baselines that apply SLSQP directly to Sobol' point sets, Fibonacci lattice, and best Phase I sets, all with the same objective and similar iteration budgets. These SLSQP baselines substantially improve over their respective seeds, but the LLM-evolved Phase II programs still achieve lower star discrepancy in 22 of the 24 tested values of $N$ with reductions of up to 15% relative to the best SLSQP-only baseline (Appendix B). This suggests the gains are not due to SLSQP alone, but to the LLM discovering nontrivial initializations and restarts that make SLSQP more effective.

## 5.2 IMPROVED DIRECTION NUMBERS

Having demonstrated the framework's ability to construct point sets, we next applied it to the discrete optimization problem of discovering improved Sobol' direction numbers. After several hundred evolutionary iterations, our LLM evolutionary search routine discovered a more performant set of parameters, focusing its modifications on the early dimensions, which are known to explain the vast majority of the variance in Asian option pricing. Specifically, the parameters for dimensions 4, 5, and 6 were updated (Appendix C). All other dimensions (up to 32) remained identical to the Joe & Kuo (2008) baseline.

To validate these new direction numbers, they were benchmarked against the standard Joe & Kuo (2008) parameters across a suite of six Asian option pricing scenarios with varying parameters (Appendix C). The true option price was computed by taking the average of 1000 randomly scrambled

Sobol sequences with $N = 2^{21}$ points each. The rQMC MSE was evaluated over 10000 random seeds for $N = 32 \ldots 8192$ points. The direction numbers discovered by LLM evolutionary search produced a significantly lower integration MSE for larger sample sizes $N \geq 512$ under one-sided Wilcoxon signed-rank test and false discovery rate correction (Table 3).

To disentangle the role of initialization from that of the LLM-guided search, we repeated the Sobol' experiment under three variants: (i) our main setup, warm-started from the Joe–Kuo parameters with Gemini Flash; (ii) the same pipeline with the smaller Gemini Flash-Lite model; and (iii) the same pipeline but initialized from random direction numbers rather than Joe–Kuo. Across six 32D Asian-style payoffs and sample sizes $N$, both Flash and Flash-Lite achieve similar MSE reductions over Joe–Kuo for $N \gtrsim 2048$, whereas the randomly initialized variant remains worse than Joe–Kuo for most $N$ (Appendix C). This suggests that the evolutionary search is leveraging a strong domain-specific prior given by Joe–Kuo, and refines it, rather than solving Sobol' design entirely from scratch.

We also compare our evolved direction numbers against Sobol' digital nets constructed by LATNET-BUILDER (L'ecuyer & Munger, 2016), using its recommended random-CBC search and projection-dependent $t$-value criterion in $d = 32$. For all six Asian-style payoffs and $N \leq 1024$, our direction numbers yield substantially lower rQMC MSE than the LatNetBuilder nets, often by factors of 2–10, with differences vanishing only for the largest $N$ where all designs converge (Appendix C).

To ensure the evolved parameters were not merely overfitted to the Asian option's specific payoff structure, we tested their generalizability on a diverse suite of high-dimensional exotic options, including Lookback, Barrier, Basket, and Bermudan options (Appendices C, D). The evolved direction numbers demonstrated strong, generalizable performance, achieving significantly lower integration error across this wider range of financial instruments for larger sample sizes ($N \geq 512$) with the sole exception of Barrier options. This suggests that the evolutionary search discovered a Sobol' sequence with fundamentally more robust and broadly applicable projection properties.

## 6    DISCUSSION

Using an LLM evolutionary framework, we generate 2D and 3D point sets with state-of-the-art star discrepancy and discover Sobol' direction numbers that lower rQMC integration error for high-dimensional exotic financial option pricing.

Recent state-of-the-art methods, such as the mathematical programming approach of Clément et al. (2024) and the GNN-based MPMC framework of Rusch et al. (2024), are designed to construct point sets for a fixed number of points $N$ and dimensions $d$. In contrast, our approach generates the direction numbers that define a Sobol' sequence, offering several key advantages:

1. **Extensibility to any $N$:** Unlike fixed approaches where a new, computationally expensive optimization must be performed from scratch for each value of $N$, a single, compact set of discovered direction numbers can be used to generate a high-quality point set for any desired number of points. This makes the solution immediately applicable to a wide range of practical problems without requiring any re-computation.

2. **Support for Progressive Integration:** An integration can be performed with $N$ points, and if more accuracy is needed, the next $N$ points from the sequence can be added to refine the estimate while reusing all previous calculations. This is a fundamental capability that static point set generation methods inherently lack.

3. **Easy Randomization:** Unlike highly optimized, deterministic point sets, the LLM optimized Sobol' sequence can make use of standard randomization techniques, such as Owen scrambling (Owen, 1995; 1998), to obtain unbiased error estimates of rQMC.

4. **High-Dimensional Applicability:** The Sobol' framework is designed from the ground up for high-dimensional integration. By optimizing the parameters within this framework, our approach is directly applicable to problems such as the 32-dimensional option pricing benchmarks used in our tests, a domain where direct coordinate optimization for a large $N$ would be computationally intractable.

Other more recent systems, such as ShinkaEvolve (Lange et al., 2025), extend the same underlying AlphaEvolve loop with additional engineering features tailored to long-horizon code benchmarks,

such as richer parent sampling, novelty filtering, and advanced orchestration of program executions. In our setting, each candidate's fitness is a cheap, deterministic quantity (star discrepancy or rQMC MSE) computed by running a short Python script, so the core evolutionary mechanism provided by OpenEvolve is sufficient to explore the search space effectively. We deliberately avoided any RL-style training or fine-tuning of the LLM: all improvements come from search in program space driven by an off-the-shelf model and the numerical fitness signal.

Taken together with the broader suite of 32D lookback, basket, and Bermudan payoffs in Appendix D, these results suggest that our evolved direction numbers are genuinely useful beyond the specific Asian option used in the fitness function: they systematically reduce MSE for a range of smooth, path-dependent integrands, but do not help (and may slightly hurt) for highly discontinuous integrands such as barrier options. We view this as an instance of problem-dependent Sobol' design: the same pipeline could in principle be re-run with an objective that mixes several payoff types to trade off specialization and robustness.

Methodologically, our work does not introduce new evolutionary operators; our contribution is empirical and conceptual: we show that a generic LLM-guided program search, applied almost out-of-the-box, is capable of rediscovering and subtly improving long-studied QMC constructions, and we analyze when these gains go beyond what can be achieved with standard local optimization alone. Empirically, our evolved Sobol' parameters work best for smooth, low-to-moderate variation integrands, and do not improve performance for highly discontinuous functions such as the payoff of barrier options.

One practical limitation of our approach is computational cost. For example, the full Sobol' search requires roughly 2000 LLM calls and on the order of $\approx 1.6 \times 10^{10}$ 32-dimensional QMC samples and payoff evaluations (Appendix A). On our CPU workstation this corresponds to about 96 wall-clock hours per Sobol' experiment. However, this cost is incurred once, offline: the resulting direction numbers can then be reused across many downstream Monte Carlo tasks.

We quantify significance using paired signed-rank tests and report all seeds; however, due to compute limits we performed one evolutionary run per problem, which we treat as a limitation. Future work could explore alternative prompting techniques, optimization of evolutionary programming hyperparameters, and meta-model methods that generate optimal direction numbers for any given dimension $d$ or point sets for any given dimension $d$ or number of points $N$.

## 7 CONCLUSION

We have demonstrated the application of an LLM-driven evolutionary framework to tackle complex discovery problems in the field of low-discrepancy sets. Our method has discovered new 2D and 3D point sets with star discrepancy values lower than any previously published, setting new benchmarks in a field of long-standing mathematical interest. Furthermore, it has produced novel Sobol' direction numbers that improve the accuracy of rQMC integration for a variety of 32-dimensional financial derivatives. This work strengthens the case for using LLMs as core components in an automated scientific discovery process, capable of generating novel and valuable mathematical knowledge.

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

> **LLM Prompt for Direct Construction**
>
> ```
> You are an expert mathematician specializing in the
> construction of QMC sampling points in a square.  Your task
> is to improve a constructor function that directly outputs
> the position of 16 points on a unit square ([0, 1] x [0, 1])
> in a way that minimizes the star discrepancy.
> The star discrepancy is a measure of how uniformly
> distributed the points are in the square.  It is defined as
> the supremum of the absolute value of the difference between
> the fraction of points and the area.
> Focus on designing an explicit constructor that specifies the
> position of each point (x, y) in the unit square, rather than
> an iterative search algorithm.
> It should output the position of each point (x, y) in the
> square [0, 1] x [0, 1].  0.0 and 1.0 are included in the
> square.
> ```

Figure 4: The full prompt provided to the LLM for generating programs that directly construct point sets with minimum star discrepancy.

## A  IMPLEMENTATION AND COMPUTATION COMPLEXITY

### A.1  INITIAL PROGRAMS

The initial parent program in 2D implemented a simple shifted Fibonacci lattice (Listing 3) and in 3D implemented a scrambled Sobol' sequence (Listing 4).

**Listing 3** Initial Program for 2D Point Set Search

```
1  def construct_star():
2      A=np.zeros((N,2))
3      phi=(math.sqrt(5)-1)/2
4      for i in range(N):
5          A[i,0]=(i+0.5)/N
6          A[i,1]=((i*phi)%1+(0.5/N))%1
7      return A
```

**Listing 4** Initial Program for 3D Point Set Search

```
1  def construct_star():
2      A=Sobol(d=3, scramble=True, seed=42).random(n=N)
3      return A
```

### A.2  LLM PROMPTS

This appendix provides the full instructional text of the prompts provided to the Large Language Model (LLM) within the OpenEvolve framework. These LLM prompts were used to generate programs to construct point sets directly (Fig. 4), iteratively optimize point sets (Fig. 5), and directly construct Sobol' direction numbers (Fig. 6).

It is important to note that the text shown in the figures below constitutes the instructional component of a larger prompt. This text is dynamically combined with the source code of a "parent" program selected for mutation and often includes code from other high-performing "inspiration" programs to encourage crossover of ideas.

---

**LLM Prompt for Iterative Optimization**

```
You are an expert mathematician specializing in the
construction of QMC sampling points in a 2D square.  Your
task is to improve a constructor function that finds the
position of 16 points on a unit square ([0, 1] x [0, 1]) in a
way that minimizes the star discrepancy.
The star discrepancy is a measure of how uniformly
distributed the points are in the square.  It is defined as
the supremum of the absolute value of the difference between
the fraction of points and the area.
Use scipy optimization routines such as
scipy.optimize.minimize to fine-tune the construction.  The
optimization routine and its initialization are critically
important.
It should output the position of each point (x, y) in the
square [0, 1] x [0, 1].  0.0 and 1.0 are included in the
square.
```

Figure 5: The full prompt provided to the LLM for generating programs that iteratively optimize point sets to have minimum star discrepancy.

---

**LLM Prompt for Sobol' Direction Numbers Search**

```
You are an expert mathematician specializing in the
construction of QMC sampling points in a square.  Your task
is to improve a constructor function that directly outputs
the direction numbers for dimensions 2 to 32 of a Sobol
Sequence.
Your goal is to minimize the approximation error of a 32
dimensional Asian option price.  The dimensions 1, 2, and 3
explain roughly 97% of the variance of the price.
The Sobol sequence is defined by a polynomial of degree s,
with coefficients represented as an integer a, and direction
numbers mᵢ for each dimension i.  The direction numbers must
be odd integers and within the specified range.
You must return a list of 31 dictionaries for directions 2 to
32, each containing the following keys:
- "s" (int):  The degree of the polynomial.  1 <= s <= 30
- "a" (int):  The coefficients of the polynomial, represented
as an integer.  0 <= a < 2ˢ⁻¹
- "mᵢ " (list[int]):  The direction numbers for the Sobol
sequence, represented as a list of integers of length s.
Each integer should be in the range [0, 2ⁱ⁺¹] and has to be
odd.
Focus on designing an explicit constructor that specifies
these parameters, rather than an iterative search algorithm.
```

Figure 6: The full prompt provided to the LLM for generating a program that directly specifies Sobol' direction numbers to have lower integration error for an Asian option.

## A.3 IMPLEMENTATION DETAILS

Our evolutionary algorithm treats each candidate solution as a full program (Python module) that implements either a point-set generator (for the 2D star-discrepancy experiments) or a Sobol' direction-

number constructor (for the QMC experiments). Each program exposes a fixed entry point with a prescribed signature (e.g., `construct_points(N)` or `construct_sobol_sequence()`), and is treated as a black box: we execute it in an isolated subprocess, validate its output, and then score it using the appropriate numerical objective.

**Population structure and archive.** We maintain a population of $P = 60$ candidate programs partitioned into four "islands" arranged in a ring topology. Each island thus holds 15 programs. The search proceeds in asynchronous iterations: at each iteration we (i) select one parent program, (ii) mutate it via an LLM call to obtain a single child, and (iii) evaluate and insert the child back into the island population. Every 25 iterations, we perform migration: one elite individual from each island is sent clockwise and replaces the worst program at the destination. In addition to the islands, we maintain a global elite archive storing the top 25 programs ever seen, ranked first by fitness and then lexicographically by (i) shorter program length and (ii) earlier discovery time. This archive is used both for parent selection and for constructing the LLM context.

**Parent selection and LLM-based mutation.** At each iteration, we choose a single parent for mutation. With probability 0.7, the parent is sampled from the elite archive; with probability 0.3, it is sampled from the union of all island populations. This biases search toward exploitation of high-performing programs while retaining exploration.

The mutation operator is a call to a Large Language Model (Google Gemini 2.0 Flash) configured as a program rewriter. Rather than applying small, local edits, we ask the LLM to produce a *complete rewrite* of the core implementation functions, conditioned on several pieces of context:

- the full source code of the parent program;

- the source code of the three best-performing programs available ("inspiration programs"), which may come from any island or from the archive;

- a system prompt specifying the required interfaces, validity constraints (e.g., dimensions, box $[0, 1]^d$, odd direction numbers), and the target objective (minimizing 2D star discrepancy or randomized QMC MSE for the Asian option).

This setup acts as a soft form of multi-parent crossover: the LLM is free to borrow subroutines and structural ideas from multiple high-quality programs but must output a single coherent child program. We use Gemini 2.0 Flash with temperature 0.7, top-$p = 0.95$, and a maximum context length of 8192 tokens. In practice, most calls use around 4000 tokens (prompt plus completion). We do not tune these LLM hyperparameters per task or perform any RL based finetuning or training on the LLM; all experiments share the same configuration.

**Program execution and validity checks.** Every newly generated program is executed in a dedicated subprocess via a small wrapper script. The wrapper:

1. imports the program as a module;

2. calls the required entry function (e.g., `construct_sobol_sequence()`);

3. serializes the returned object to a temporary file.

The main process then deserializes this object and performs a strict validity check. For Sobol' direction-number search, we require: 32 total dimensions; each dimension entry to contain fields $s$, $a$, and $m$; $1 \leq s \leq 30$; $0 \leq a < 2^{s-1}$; $|m| = s$; each direction number satisfies $1 \leq m_i < 2^s$ and is odd.

If any of these checks fails, if the program crashes, or if execution exceeds a 600 second wall-clock timeout, the candidate is marked invalid and assigned zero fitness. For the 2D star discrepancy experiments, we analogously check that the program returns an $N$-point set in $[0, 1]^2$ with the correct shape and bounds; a wall-clock timeout of 600 seconds is used.

**Fitness evaluation.** For valid programs, we compute fitness using the same numerical objectives as in the main experiments:

- **2D star discrepancy.** Given an $N$-point set $P \subset [0,1]^2$ returned by the program, we compute the star discrepancy $D^*(P)$ and define fitness as a monotone decreasing function of $D^*(P)$: $f = \frac{1}{1+D^*(P)}$. This preserves the ranking of candidate programs while providing a scalar objective for evolution.

- **Sobol' direction numbers (randomized QMC MSE).** For Sobol', the program outputs a list of 31 parameter triples $(s, a, \{m_i\})$ defining dimensions $2, \ldots, 32$. To score a candidate $A$, we estimate the mean squared error (MSE) of a randomized QMC estimator for the 32-dimensional Asian call option used throughout our QMC experiments. Each evaluation draws $R = 1000$ independent randomizations (digital shifts and lower-triangular scrambling matrices) and, for each randomization, generates $N = 8192$ Sobol' points in $d = 32$ via a compiled C++ generator. For each point set we evaluate the Asian payoff and price estimator $\hat{C}$, accumulate the squared errors $(\hat{C} - C_0)^2$ relative to the known ground-truth price $C_0$, and define

$$f = \frac{1}{1 + \sum_{r=1}^{R}(\hat{C}^{(r)} - C_0)^2}$$

**Search budget and hardware.** For each task (2D star discrepancy and Sobol' direction numbers), we run at most 2000 evolutionary iterations, each evaluating exactly one new candidate program. Including the initial population, this yields at most a few thousand distinct programs per run. In practice, a nontrivial fraction of LLM-generated programs fail validation (e.g., due to syntax errors or invalid solutions) and are quickly discarded with zero fitness; the bulk of computation is spent on valid candidates that survive to be evaluated. All numerical evaluations (star discrepancy, Sobol' generation, QMC payoffs) are executed on CPU. Experiments were carried out on a workstation with an AMD EPYC 7763 CPU and 64 GB of RAM. A full Sobol' direction-number run at this budget took approximately 96 hours of wall-clock time. We did not perform any hyperparameter tuning of the evolutionary algorithm; population size, archive size, migration schedule, parent selection probabilities, and LLM parameters are fixed across all experiments. Code and all generated point sets and Sobol' parameters are available at `https://github.com/hockeyguy123/openevolve-star-discrepancy`.

A.4 COMPUTATIONAL COMPLEXITY

The total computational cost of our method has three main components: (i) LLM-based mutation; (ii) program execution and validation; and (iii) numerical fitness evaluation. We analyze each in turn.

**LLM mutation.** Let $T_{\text{prompt}}$ and $T_{\text{output}}$ denote the number of tokens in the input and output of one LLM call. Each mutation prompt consists of: a fixed system instruction block, the source code of the parent program, and the source of three top "inspiration" programs. If the typical program size is $L$ lines of code, then both the prompt and the completion are $\mathcal{O}(L)$ in length, so that one mutation costs $\mathcal{O}(T_{\text{prompt}} + T_{\text{output}}) = \mathcal{O}(L)$ tokens. With at most $C_{\text{LLM}} = 2000$ LLM calls per run, the total LLM complexity per run is

$$\mathcal{O}(C_{\text{LLM}} \cdot L).$$

In practice, we cap the context length at 8192 tokens, while observed calls average around 4000 tokens (prompt plus completion), so the token budget scales linearly with the number of iterations.

**Program execution and validation.** Program execution cost per candidate scales with program length and the cost of constructing its output. Under our design, programs generate their outputs symbolically (using explicit formulas and loops) rather than by performing expensive internal sampling. For the tasks considered:

- in 2D star-discrepancy experiments, point generation costs $\mathcal{O}(N)$ per program;
- in Sobol' direction-number experiments, the program simply returns a list of 31 parameter triples, which is $\mathcal{O}(1)$.

Validity checks for Sobol' parameters are linear in the size of this list (constant with respect to $N$), and validity checks for 2D point sets are linear in the number of points $N$. Compared to fitness evaluation, these costs are negligible.

**Numerical fitness evaluation.** The dominant cost lies in computing the numerical objective for each valid program.

- **2D star discrepancy.** For a point set $P$ with $N$ points in $d = 2, 3$, we use a star discrepancy routine that scales exponentially with dimension $\mathcal{O}(N^{d/2+1})$. Let $C_{\mathrm{disc}}(N)$ denote this cost. If at most $K$ distinct programs are evaluated in one run, the total cost for star discrepancy is
$$\mathcal{O}\big(K \cdot C_{\mathrm{disc}}(N)\big).$$
For our ranges of $N$ and $d$, this term is modest relative to the Sobol' QMC experiments.

- **Sobol' direction numbers (QMC MSE).** For Sobol', a single fitness evaluation of a candidate requires $R = 1000$ randomized QMC estimators, each with $N = 8192$ points and dimension $d = 32$. The compiled C++ Sobol' generator produces $N$ points in $\mathcal{O}(Nd)$ time per randomization, and the Asian payoff evaluation likewise costs $\mathcal{O}(Nd)$: each path requires a cumulative sum over $d$ steps, exponentials, and a payoff computation. Thus one fitness evaluation costs
$$\mathcal{O}\big(R \cdot N \cdot d\big).$$
With $R = 1000$, $N = 8192$, and $d = 32$, each evaluation processes $R \times N \approx 8.2 \times 10^6$ 32-dimensional points, corresponding to on the order of $2.6 \times 10^8$ scalar operations. If at most $K \leq 2000$ distinct Sobol' programs are evaluated, the total numerical cost per run is
$$\mathcal{O}\big(K \cdot R \cdot N \cdot d\big),$$
which in our setting corresponds to roughly $K \times R \times N \approx 1.6 \times 10^{10}$ 32-dimensional QMC samples and Asian payoffs per full evolutionary run.

**Overall complexity.** Combining the above, the total computational complexity per run can be summarized as
$$\underbrace{\mathcal{O}\big(C_{\mathrm{LLM}} \cdot L\big)}_{\text{LLM mutations}} + \underbrace{\mathcal{O}(K)}_{\text{program execution / validation}} + \underbrace{\mathcal{O}\big(K \cdot C_{\mathrm{fit}}\big)}_{\text{fitness evaluations}},$$
where $C_{\mathrm{fit}} = C_{\mathrm{disc}}(N)$ for 2D discrepancy and $C_{\mathrm{fit}} = RNd$ for Sobol'. In all our Sobol' experiments the $KRNd$ term dominates, while for 2D discrepancy the LLM term and the discrepancy term are of comparable size. Crucially, this entire cost is incurred once, *offline*. Using the resulting point sets or Sobol' parameters in downstream applications has the same asymptotic cost as using a standard hand-designed formula: generating $N$ Sobol' points is still $\mathcal{O}(Nd)$, and evaluating QMC estimators with them is unchanged.

## B  GENERATED POINT SETS

This appendix provides the full numerical results for the star discrepancy ($D_N^*$) values of the point sets discovered by LLM evolutionary search. These tables serve as a comprehensive record of the performance of our method, establishing new state-of-the-art benchmarks and offering a detailed comparison against established low-discrepancy construction methods.

### B.1  STAR DISCREPANCY EVALUATION

We evaluate $D_N^*$ by enumerating all anchored boxes whose upper corners lie on the Cartesian product of per-dimension grids $G_j = \{\text{unique } x_{i,j}\} \cup \{1\}$, with the lower corner fixed at the origin. For each candidate corner $\mathbf{y} \in G_1 \times \cdots \times G_d$, we compute its volume $\prod_j y_j$ and two point counts: the number of samples dominated by $\mathbf{y}$ (component-wise $\leq$) and the same count with points on any upper face removed. The local discrepancy is the maximum of the two absolute differences from the volume, capturing the half-open convention on anchored boxes. Inputs are clipped to $[0, 1]^d$, grids are built from unique coordinates (plus the terminal 1.0).

### B.2  NEW BENCHMARKS IN THREE DIMENSIONS

We report high-quality constructions for $N$ from 9 to 16 in three dimensions (Table 4).

| N | LLM Discrepancy ($D_N^*$) |
|---|---|
| 9 | 0.1758 |
| 10 | 0.1652 |
| 11 | 0.1551 |
| 12 | 0.1483 |
| 13 | 0.1402 |
| 14 | 0.1337 |
| 15 | 0.1275 |
| 16 | 0.1207 |

Table 4: The Star Discrepancy Values for $N > 8$ 3D point sets found via LLM evolutionary search.

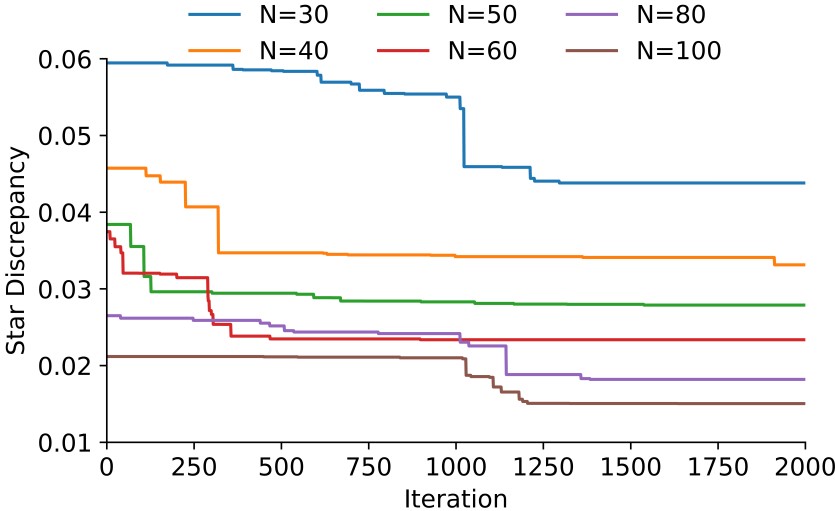

Figure 7: Best-so-far star discrepancy in 2D for $N \in \{30, 40, 50, 60, 80, 100\}$ as a function of iteration. Each curve corresponds to a single evolutionary run; the value at iteration $t$ is the minimum discrepancy observed up to that point.

### B.3 OPTIMIZATION TRAJECTORIES IN 2D

To better understand how the evolutionary loop improves the point sets over time, we record the best-so-far star discrepancy after each iteration for several values of $N$ in 2D. Figure 7 shows the resulting trajectories for $N \in \{30, 40, 50, 60, 80, 100\}$ over the full 2000-iteration budget.

Across all $N$, we observe a characteristic two-phase behavior. First, the discrepancy improves rapidly during the early iterations, roughly the first 200–400 steps, dropping sharply from the initial constructions.

After this initial phase, the curves enter a slower regime of incremental refinement. They continue to decrease, but with smaller step sizes and less frequent improvements. Occasional late "jumps"—for example, for $N = 80$ around iteration 1100 and for $N = 30$ around iteration 1200—show that useful innovations can still be discovered well into the run. However, these gains are modest compared to those obtained early on.

### B.4 EXTENDED PERFORMANCE IN TWO DIMENSIONS

In two dimensions, while the problem is better understood than in 3D, optimal solutions for larger point sets ($N > 21$) are not known. We provide a detailed comparison of the 2D star discrepancy values achieved by LLM evolutionary optimization against several key baselines for $N$ ranging from 140 to 1020 (Table 5).

| N | Halton | Sobol' | Hammersley | Fibonacci | MPMC | LLM |
|---|--------|--------|------------|-----------|------|-----|
| 140 | 0.03686 | 0.02794 | 0.02701 | 0.01870 | 0.01373 | **0.01151** |
| 180 | 0.03200 | 0.02300 | 0.02283 | 0.01454 | 0.01147 | **0.01058** |
| 220 | 0.02323 | 0.01924 | 0.01868 | 0.01190 | 0.00975 | **0.00891** |
| 260 | 0.02062 | 0.01546 | 0.01581 | 0.01063 | 0.00843 | **0.00831** |
| 300 | 0.01994 | 0.01341 | 0.01424 | 0.00972 | 0.00752 | **0.00741** |
| 340 | 0.01950 | 0.01353 | 0.01266 | 0.00858 | 0.00710 | **0.00696** |
| 380 | 0.01659 | 0.01167 | 0.01170 | 0.00768 | 0.00695 | **0.00638** |
| 420 | 0.01617 | 0.01194 | 0.01058 | 0.00694 | 0.00584 | **0.00563** |
| 460 | 0.01279 | 0.00972 | 0.00966 | 0.00634 | 0.00540 | **0.00471** |
| 500 | 0.01172 | 0.00848 | 0.00889 | 0.00583 | 0.00518 | **0.00509** |
| 540 | 0.01101 | 0.00967 | 0.00823 | 0.00540 | 0.00488 | **0.00407** |
| 580 | 0.01261 | 0.00956 | 0.00766 | 0.00503 | 0.00476 | **0.00377** |
| 620 | 0.01140 | 0.00782 | 0.00744 | 0.00470 | 0.00454 | **0.00386** |
| 660 | 0.01074 | 0.00956 | 0.00699 | 0.00463 | 0.00437 | **0.00360** |
| 700 | 0.00933 | 0.00865 | 0.00682 | 0.00437 | 0.00416 | **0.00379** |
| 740 | 0.00891 | 0.00709 | 0.00646 | 0.00435 | 0.00397 | **0.00346** |
| 780 | 0.00870 | 0.00662 | 0.00612 | 0.00412 | 0.00376 | **0.00310** |
| 820 | 0.00881 | 0.00678 | 0.00583 | 0.00392 | 0.00360 | **0.00294** |
| 860 | 0.00914 | 0.00604 | 0.00556 | 0.00374 | 0.00356 | **0.00316** |
| 900 | 0.00751 | 0.00561 | 0.00531 | 0.00357 | 0.00319 | **0.00296** |
| 940 | 0.00785 | 0.00481 | 0.00508 | 0.00342 | 0.00321 | **0.00289** |
| 980 | 0.00768 | 0.00558 | 0.00487 | 0.00328 | 0.00308 | **0.00280** |
| 1020 | 0.00777 | 0.00395 | 0.00468 | 0.00315 | 0.00303 | **0.00245** |

Table 5: 2D Star Discrepancy Comparison for $N \geq 140$. This table compares the performance of LLM evolutionary search against classical low-discrepancy sequences and the state-of-the-art MPMC method for larger point sets where optimal solutions are not known. Lower values indicate better uniformity. The best result in each row is **bolded**.

- **Consistent Outperformance:** Across every single value of $N$ tested, the point sets discovered by LLM evolutionary search achieve a lower star discrepancy than all other methods, including classical sequences (Halton, Sobol', Hammersley, Fibonacci) and the recent state-of-the-art MPMC method.

- **Significant Improvement Margin:** The performance gap is not trivial. For example, at $N = 140$, the LLM discrepancy of 0.01151 is approximately 16% lower than the next-best method (MPMC at 0.01373) and over 57% lower than the widely used Halton sequence. At $N = 1020$, the LLM discrepancy of 0.00245 is nearly 20% better than MPMC's 0.00303.

- **Superior Scaling:** The results demonstrate that the LLM evolutionary search's ability to find superior configurations is not limited to a specific range of $N$, but holds consistently as the number of points increases. This suggests that the two-phase discovery strategy (direct construction followed by iterative optimization) is effective at navigating the increasingly complex search space associated with larger point sets.

## B.5 PHASE-II LOCAL OPTIMIZATION

To determine whether Phase II improvements could be attributable solely to the use of a standard local optimizer, we constructed a suite of baselines in 2D that used purely local optimization, specifically sequential least squares programming (SLSQP) which is the optimizer predominantly used by the LLM constructed programs. For each $N \in \{2, \ldots, 20, 30, 40, 50, 60, 100\}$ we consider three initial point sets: the Sobol' sequence, a simple Fibonacci construction, and the best point set returned by the LLM's Phase I constructive search. From each of these initializations, we then run a single SLSQP local optimization directly on the point coordinates, using the same star-discrepancy objective as in the main experiments and box constraints $[0, 1]^2$ for all points. The resulting sets are denoted *Sobol+SLSQP*, *Fibonacci+SLSQP*, and *Phase 1+SLSQP*. Importantly, these baselines use the same objective and a comparable iteration budget to Phase II, but without any LLM-designed program structure (no LLM-chosen restarts, parameterizations, or heuristics). For comparison, we also report the LLM-evolved point sets from Phase II ("LLM").

| N | Sobol | Sobol SLSQP | Fibonacci | Fibonacci SLSQP | Phase I | Phase I SLSQP | LLM |
|---|---|---|---|---|---|---|---|
| 2 | 0.7500 | **0.3660** | 0.5365 | **0.3660** | 0.4375 | **0.3660** | **0.3660** |
| 3 | 0.6250 | 0.3333 | 0.4850 | 0.3333 | 0.3375 | 0.3333 | **0.2847** |
| 4 | 0.4375 | **0.2500** | 0.3637 | **0.2500** | 0.2856 | **0.2500** | **0.2500** |
| 5 | 0.4375 | **0.2000** | 0.2910 | **0.2000** | 0.2500 | 0.2500 | **0.2000** |
| 6 | 0.2917 | 0.1833 | 0.2836 | 0.1795 | 0.2078 | 0.1771 | **0.1667** |
| 7 | 0.3393 | 0.1571 | 0.2431 | 0.1587 | 0.1870 | 0.1518 | **0.1500** |
| 8 | 0.3125 | 0.1495 | 0.2127 | 0.1389 | 0.1697 | 0.1429 | **0.1328** |
| 9 | 0.3264 | 0.1318 | 0.1891 | 0.1270 | 0.1551 | 0.1270 | **0.1235** |
| 10 | 0.2906 | 0.1318 | 0.1702 | 0.1143 | 0.1375 | 0.1172 | **0.1111** |
| 11 | 0.2088 | 0.1099 | 0.1547 | 0.1074 | 0.1255 | 0.1095 | **0.1039** |
| 12 | 0.2240 | 0.1303 | 0.1418 | 0.0985 | 0.1258 | 0.0979 | **0.0960** |
| 13 | 0.2404 | 0.1176 | 0.1309 | 0.0909 | 0.1161 | 0.0918 | **0.0892** |
| 14 | 0.1970 | 0.1092 | 0.1295 | 0.0850 | 0.1075 | 0.0857 | **0.0844** |
| 15 | 0.2094 | 0.1020 | 0.1209 | 0.0793 | 0.1012 | 0.0804 | **0.0791** |
| 16 | 0.1719 | 0.0945 | 0.1264 | **0.0743** | 0.0962 | 0.0757 | 0.0745 |
| 17 | 0.1976 | 0.0937 | 0.1190 | 0.0732 | 0.0903 | 0.0735 | **0.0712** |
| 18 | 0.1649 | 0.0980 | 0.1124 | 0.0691 | 0.0885 | **0.0676** | 0.0676 |
| 19 | 0.1357 | 0.0969 | 0.1065 | 0.0673 | 0.0850 | 0.0665 | **0.0654** |
| 20 | 0.1313 | 0.0824 | 0.1011 | 0.0642 | 0.0818 | 0.0633 | **0.0611** |
| 30 | 0.0690 | 0.0613 | 0.0674 | 0.0442 | 0.0595 | 0.0440 | **0.0438** |
| 40 | 0.0836 | 0.0481 | 0.0579 | 0.0356 | 0.0457 | 0.0365 | **0.0331** |
| 50 | 0.0745 | 0.0379 | 0.0463 | 0.0281 | 0.0355 | **0.0278** | 0.0278 |
| 60 | 0.0484 | 0.0349 | 0.0386 | 0.0236 | 0.0320 | 0.0253 | **0.0234** |
| 100 | 0.0423 | 0.0227 | 0.0245 | 0.0153 | 0.0212 | 0.0161 | **0.0150** |

Table 6: 2D star discrepancy comparison for $N \in \{2, \ldots, 20, 30, 40, 50, 60, 100\}$ between Sobol, Fibonacci, Phase 1 (with and without SLSQP refinement), and LLM search. Best values are **bolded**.

Table 6 reports the resulting star discrepancies. As expected, adding SLSQP on top of classical constructions already yields a large improvement over the Sobol, Fibonacci, and Phase 1 sets, confirming that local optimization is a strong baseline for this problem. However, the LLM-evolved Phase II sets consistently achieve even lower star discrepancies. In 22 out of the 24 values of $N$ in Table 6, the LLM achieved strictly lower star discrepancy than every SLSQP baseline (Sobol+SLSQP, Fibonacci+SLSQP, Phase 1+SLSQP). The largest relative improvements occur at small and moderate $N$ (e.g., $0.3333 \rightarrow 0.2847$ at $N = 3$, a reduction of about $15\%$, and $0.0356 \rightarrow 0.0331$ at $N = 40$, roughly $7\%$). In the remaining two cases ($N = 16$ and $N = 50$), the best SciPy baseline is better by less than $2 \times 10^{-4}$ in absolute discrepancy.

These results show that while local optimization is essential for obtaining strong point sets, the LLM-guided Phase II search still provides a measurable advantage over just local optimization starting from strong human-designed seeds.

### B.6 PROMPTING VS EVOLUTIONARY OPTIMIZATION AND LLM SIZE

| Method | Mean | Std. dev. | Min | Max |
|---|---|---|---|---|
| singleturn | 0.02117 | 0.00012 | 0.02090 | 0.02135 |
| multiturn | 0.01852 | 0.00254 | 0.01479 | 0.02098 |
| LLM | 0.01541 | 0.00065 | 0.01492 | 0.01756 |
| LLM-lite | 0.01540 | 0.00064 | 0.01492 | 0.01690 |

Table 7: Ablation at $N = 100$ (2D): star discrepancy over 16 seeds for single-shot prompting (`singleturn`), multi-turn prompting without evolution (`multiturn`), and evolutionary search with the Gemini Flash model (`LLM`) or the smaller Flash-Lite model (`LLM-lite`).

We next isolate the contributions of (i) the evolutionary search loop versus plain iterative prompting, and (ii) the choice of LLM. At $N = 100$ in 2D, we ran 16 independent seeds for four variants:

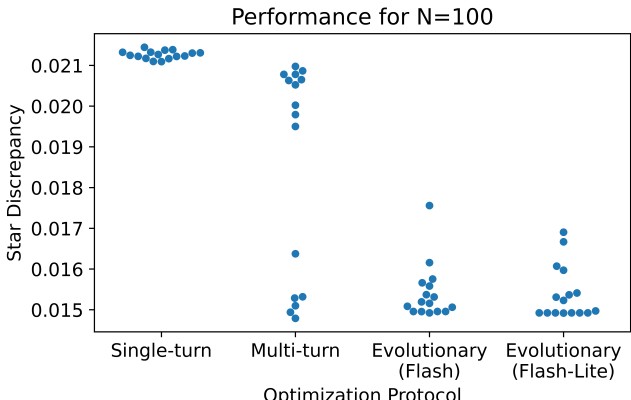

Figure 8: Per-seed star discrepancy at $N = 100$ in 2D for single-turn and multi-turn prompting, and evolutionary search with Gemini Flash and Flash-Lite. Each point is one run with a different seed.

- **singleturn**: a single one-shot call to the main LLM (no iteration, no evolution);

- **multiturn**: repeated prompting of the LLM for up to 2000 calls, but without evolutionary operators (no population, no selection, no mutation); this is an iterative prompting baseline;

- **LLM**: the full evolutionary pipeline with the main Gemini Flash model, including population-based evolutionary search (selection, mutation, and archiving);

- **LLM-lite**: the same evolutionary pipeline as `LLM`, but using a smaller Flash-Lite model as the LLM.

All variants optimize the same 2D star-discrepancy objective as in the main experiments. Table 7 summarizes the distribution of the best star discrepancy for $N = 100$ points over the 16 seeds for each method, and Figure 8 visualizes the per-seed outcomes.

First, the single-shot baseline is insufficient to reach the low-discrepancy regime we target. Second, simply calling the LLM multiple times without an evolutionary mechanism (`multiturn`) substantially improves over single-turn prompting (mean 0.0185 vs. 0.0212) and occasionally discovers very strong solutions (best seed 0.01479). However, this baseline exhibits high variability across seeds (standard deviation $\approx 2.5 \times 10^{-3}$), with some runs remaining close to the single-turn performance. This variability is clearly visible in Figure 8.

Third, the full evolutionary search variants (`LLM` and `LLM-lite`) achieve consistently lower discrepancies on average ($\approx 0.0154$) with much smaller variance ($\approx 6.5 \times 10^{-4}$). Their best runs (0.01492) are within $1.3 \times 10^{-4}$ of the best `multiturn` run, but the evolutionary runs are far more robust: almost all seeds converge to strong solutions, as seen in the tight clusters on the right of Figure 8. The comparison between `LLM` and `LLM-lite` further indicates that the overall evolutionary search pipeline remains effective even when replacing the main LLM with a smaller Flash-Lite model; their means and variances are nearly identical.

## C  GENERATED SOBOL' DIRECTION NUMBERS

| D | S | A | $M_i$ |
|---|---|---|---|
| 4 | 3 | 1 | 1 3 5 |
| 5 | 3 | 2 | 1 3 7 |
| 6 | 4 | 1 | 1 1 3 7 |

Table 8: Sobol' direction number parameters updated by LLM evolutionary search. D is the dimension, S is the polynomial degree, A is the polynomial's coefficients, and $M_i$ are the initial direction numbers. All other dimensions remained unchanged.

We provide the direction numbers discovered by the LLM evolutionary search routine. Only the parameters for dimensions 4, 5, and 6 were updated (Table 8).

## C.1   COMPARISON WITH LATNETBUILDER DIGITAL NETS

We compare Sobol' direction numbers found via evolutionary LM search against digital nets constructed with LATNETBUILDER, a state-of-the-art QMC design tool. We use the recommended random-CBC search (2000 iterations) in LatNetBuilder with a projection-dependent $t$-value based criterion in $d = 32$ dimensions for $2^N$ points:

```
latnetbuilder -t net -c sobol -s 2^N -d 32 \
  -e random-CBC:2000 -f projdep:t-value -q inf \
  -w order-dependent:0:0,1,1
```

For each resulting Sobol digital net and each number of points $N \in \{32, 64, \ldots, 8192\}$, we construct randomized QMC estimators for the six 32-dimensional Asian-style option-pricing payoffs considered in the training example, out-of-the-money, at-the-money, in-the-money, high volatility, and low volatility. Table 9 reports, for each payoff and $N$, the empirical MSE for LatNetBuilder vs. our implementation (LLM), together with a p-value from a one-sided Wilcoxon signed-rank test with false discovery rate correction across payoffs for each $N$; significant p-values ($p < 0.05$) are bolded.

For small to moderate sample sizes ($N \leq 1024$), the LLM discovered Sobol' sequence consistently and substantially outperforms the LatNetBuilder Sobol' digital nets, often reducing MSE by factors between $2\times$ and $10\times$, with extremely small $p$-values across all scenarios. As $N$ increases further, the MSEs of the two constructions converge. In this regime differences are small in absolute terms and typically not statistically significant. Overall, the Sobol' direction numbers found by our implementation are competitive with, and usually improves upon, Sobol' direction numbers optimized by a dedicated QMC design tool over a wide range of sample sizes.

## C.2   ROLE OF LLM SIZE AND INITIALIZATION FOR SOBOL' DIRECTION NUMBERS

To better understand the role of LLM size and the importance of initialization in our direction-number search, we compare three LLM variants against the standard Joe–Kuo Sobol' sequence on the six 32D Asian-style options used in our main randomized QMC experiments. For each variant, we run 16 independent evolutionary searches with different random seeds and report the mean $\pm$ standard deviation of the resulting randomized QMC MSE. The direction numbers used for evaluation are exactly those optimized on the training example; they are not retuned for the other payoffs. The results are summarized in Table 10.

The first two variants differ only in the underlying LLM: LLM uses Gemini Flash, while LLM-lite replaces it with Gemini's smaller Flash-Lite model. Across all scenarios and sample sizes $N$, their MSEs are very similar, with differences typically well within one standard deviation. On the training example, for instance, both variants have slightly worse MSE than Joe–Kuo for small $N$ but achieve lower MSE for larger $N \geq 2048$. A similar pattern holds for the out-of-the-money, at-the-money, and volatility-shifted payoffs: for $N \gtrsim 2,000$, both LLM-based variants match or modestly improve on the Joe–Kuo baseline across almost all scenarios, despite being optimized only on the training payoff. This suggests that the overall evolutionary pipeline is not overly sensitive to LLM size; a smaller model can still discover high-quality direction numbers. Furthermore, both LLM and LLM-lite have relatively low standard deviations (typically less than 25% of the mean for large $N$), reflecting their stability across evolutionary runs.

The third variant, LLM (random init), replaces the Joe–Kuo direction numbers with a random initialization in the same parameter space, keeping the rest of the OpenEvolve pipeline unchanged. This variant is consistently worse than standard Sobol' at small and moderate $N$ (often by factors of $2\times$–$5\times$ in MSE), but approaches baseline performance at the largest $N$ (e.g., $N = 8192$). In several scenarios, such as at-the-money and high volatility, it remains worse than Joe–Kuo. These results indicate that, under a realistic compute budget, evolutionary search alone is not sufficient to recover high-quality Sobol' parameters from scratch; a strong initialization such as Joe–Kuo is a crucial inductive bias.

# D    OPTION SCENARIOS

To ensure a rigorous evaluation of the Sobol' direction numbers discovered, we designed a comprehensive suite of benchmark scenarios. This suite includes the primary optimization target (the Asian option, see Table 11) as well as a diverse set of exotic options known to be challenging for Quasi-Monte Carlo integration. The purpose of this suite is twofold: first, to confirm superior performance on the target problem class, and second, to test for generalizability and ensure that the evolved parameters were not merely overfitted to the specific payoff structure of the Asian option.

The configuration parameters for all tested scenarios are consolidated in Table 12. For all options, the time to expiration (T) was set to 1.0 year and the risk-free interest rate (r) was 0.05. The underlying asset prices are assumed to follow a geometric Brownian motion.

## D.1    ASIAN OPTIONS

An Asian option is a path-dependent exotic option whose payoff is determined by the average price of the underlying asset over a pre-set period of time. This is in contrast to a standard European option, which only depends on the asset price at expiration. The averaging feature reduces volatility and makes the option generally cheaper than its European counterpart. Because its value depends on the entire price path, pricing it requires simulating all 32 time steps, making it an excellent candidate for QMC methods.

## D.2    LOOKBACK OPTIONS

A lookback option is another path-dependent option whose payoff is determined by the maximum or minimum price of the underlying asset over the option's life. The scenarios tested here are for a floating strike lookback call option, whose payoff at expiration is the difference between the final asset price and the minimum price achieved ($S_T - S_{min}$).

## D.3    BARRIER OPTIONS

A barrier option is a path-dependent option that is either activated ("knocks-in") or extinguished ("knocks-out") if the underlying asset price crosses a predetermined "barrier" level. This feature introduces a significant discontinuity in the payoff function.

## D.4    BASKET OPTIONS

A basket option's payoff depends on the value of a portfolio or "basket" of multiple underlying assets. It is an inherently high-dimensional problem where the correlation ($\rho$) between the assets is a critical parameter. For these tests, we assume a uniform initial price of $S_0 = 100.00$ for all 32 assets in the basket.

## D.5    BERMUDAN OPTIONS

A Bermudan option is a hybrid between a European option (exercisable only at expiration) and an American option (exercisable at any time). It can be exercised on a discrete set of pre-specified dates. Pricing a Bermudan option is a highly complex problem that requires solving a dynamic programming problem to determine the optimal exercise strategy.

# E    RQMC INTEGRATION RESULTS

This appendix provides the detailed results for the primary benchmark (Asian option) and the generalizability tests (exotic options). Each table compares the performance of the standard Sobol' sequence (Joe & Kuo) against the direction numbers discovered by LLM evolutionary search. We report the Mean Squared Error (MSE) and its constituent parts, Squared Bias and Variance, for an increasing number of points (N). We consistently found lower variance and MSE for all option types for all $N \geq 512$ with two exceptions: close barrier option ($N = 2048$) and a high correlation basket option ($N = 2048$).

| Scenario | $N$ | LatNet MSE | LLM MSE | p-value |
|---|---|---|---|---|
| Training Example | 32 | 3.472e-01 | **2.523e-01** | **1.97e-44** |
| | 64 | 1.308e-01 | **6.649e-02** | **1.55e-177** |
| | 128 | 7.057e-02 | **1.922e-02** | **0.00e+00** |
| | 256 | 1.809e-02 | **6.108e-03** | **0.00e+00** |
| | 512 | 1.384e-02 | **1.614e-03** | **0.00e+00** |
| | 1024 | 9.989e-03 | **5.244e-04** | **0.00e+00** |
| | 2048 | 2.753e-04 | **2.254e-04** | **4.15e-15** |
| | 4096 | 1.213e-04 | **9.346e-05** | **9.37e-27** |
| | 8192 | 4.286e-05 | **4.104e-05** | 1.63e-01 |
| Out-of-the-Money | 32 | 2.532e-01 | **1.979e-01** | **7.18e-25** |
| | 64 | 1.002e-01 | **6.309e-02** | **6.55e-77** |
| | 128 | 5.247e-02 | **2.313e-02** | **8.85e-261** |
| | 256 | 1.597e-02 | **8.893e-03** | **1.46e-148** |
| | 512 | 1.087e-02 | **3.311e-03** | **0.00e+00** |
| | 1024 | 6.971e-03 | **1.313e-03** | **0.00e+00** |
| | 2048 | 5.971e-04 | **5.353e-04** | **3.29e-07** |
| | 4096 | 2.459e-04 | **2.432e-04** | 1.77e-01 |
| | 8192 | **8.641e-05** | 1.066e-04 | 1.00e+00 |
| At-the-Money | 32 | 4.503e-01 | **3.346e-01** | **1.46e-42** |
| | 64 | 1.691e-01 | **9.079e-02** | **1.74e-152** |
| | 128 | 9.224e-02 | **2.822e-02** | **0.00e+00** |
| | 256 | 2.413e-02 | **9.815e-03** | **0.00e+00** |
| | 512 | 1.794e-02 | **3.106e-03** | **0.00e+00** |
| | 1024 | 1.269e-02 | **1.168e-03** | **0.00e+00** |
| | 2048 | 5.418e-04 | **5.212e-04** | 3.04e-01 |
| | 4096 | 2.355e-04 | **2.261e-04** | 1.18e-01 |
| | 8192 | **9.411e-05** | 9.523e-05 | 1.00e+00 |
| In-the-Money | 32 | 1.925e-01 | **1.428e-01** | **1.95e-37** |
| | 64 | 7.452e-02 | **3.818e-02** | **2.27e-168** |
| | 128 | 3.849e-02 | **1.110e-02** | **0.00e+00** |
| | 256 | 1.037e-02 | **3.373e-03** | **0.00e+00** |
| | 512 | 7.949e-03 | **8.593e-04** | **0.00e+00** |
| | 1024 | 5.697e-03 | **2.601e-04** | **0.00e+00** |
| | 2048 | 1.315e-04 | **1.000e-04** | **2.91e-33** |
| | 4096 | 6.003e-05 | **3.949e-05** | **2.16e-72** |
| | 8192 | 1.693e-05 | **1.683e-05** | 6.02e-01 |
| High Volatility | 32 | 2.902e+00 | **2.149e+00** | **1.25e-40** |
| | 64 | 1.133e+00 | **5.958e-01** | **1.74e-152** |
| | 128 | 6.053e-01 | **1.877e-01** | **0.00e+00** |
| | 256 | 1.647e-01 | **6.425e-02** | **0.00e+00** |
| | 512 | 1.235e-01 | **1.959e-02** | **0.00e+00** |
| | 1024 | 8.672e-02 | **7.060e-03** | **0.00e+00** |
| | 2048 | 3.225e-03 | **2.867e-03** | **1.75e-04** |
| | 4096 | 1.365e-03 | **1.233e-03** | **4.04e-05** |
| | 8192 | **4.255e-04** | 5.321e-04 | 1.00e+00 |
| Low Volatility | 32 | 3.253e-02 | **2.455e-02** | **2.93e-38** |
| | 64 | 1.200e-02 | **6.833e-03** | **2.93e-129** |
| | 128 | 6.584e-03 | **2.205e-03** | **0.00e+00** |
| | 256 | 1.726e-03 | **8.026e-04** | **6.07e-252** |
| | 512 | 1.251e-03 | **2.747e-04** | **0.00e+00** |
| | 1024 | 8.675e-04 | **1.088e-04** | **0.00e+00** |
| | 2048 | 5.024e-05 | **4.961e-05** | 5.67e-01 |
| | 4096 | 2.207e-05 | **2.196e-05** | 6.46e-01 |
| | 8192 | 9.317e-06 | **9.300e-06** | 1.00e+00 |

Table 9: Mean-squared error (MSE) of randomized QMC estimators for six 32-dimensional Asian-style option payoffs, comparing digital nets constructed with LATNETBUILDER to those optimized by LLM evolutionary search, for $N \in \{32, 64, \ldots, 8192\}$. Lowest MSE in each row is **bolded**. The last column reports FDR-corrected one-sided Wilcoxon p-values; significant differences ($p < 0.05$) are **bolded**.

| Scenario | $N$ | Sobol MSE | LLM MSE | LLM-lite MSE | LLM MSE Random Init |
|---|---|---|---|---|---|
| Training | 32 | 2.48e-01 | 3.12e-01 ± 1.48e-01 | 3.17e-01 ± 9.02e-02 | 6.20e-01 ± 1.81e-01 |
| Example | 64 | 6.42e-02 | 1.05e-01 ± 8.77e-02 | 1.11e-01 ± 5.53e-02 | 2.85e-01 ± 1.04e-01 |
| | 128 | 1.83e-02 | 3.39e-02 ± 3.80e-02 | 3.75e-02 ± 2.53e-02 | 1.34e-01 ± 5.56e-02 |
| | 256 | 5.55e-03 | 1.26e-02 ± 1.85e-02 | 1.32e-02 ± 8.09e-03 | 5.90e-02 ± 2.24e-02 |
| | 512 | 1.65e-03 | 4.22e-03 ± 6.52e-03 | 2.64e-03 ± 1.63e-03 | 1.95e-02 ± 9.69e-03 |
| | 1024 | 5.42e-04 | 1.66e-03 ± 3.39e-03 | 7.97e-04 ± 6.39e-04 | 7.28e-03 ± 4.31e-03 |
| | 2048 | 2.33e-04 | 2.26e-04 ± 3.91e-06 | 2.43e-04 ± 1.65e-05 | 2.52e-03 ± 2.75e-03 |
| | 4096 | 9.88e-05 | 9.72e-05 ± 3.02e-06 | 1.01e-04 ± 7.70e-06 | 3.49e-04 ± 4.03e-04 |
| | 8192 | 4.52e-05 | 4.18e-05 ± 1.14e-06 | 4.25e-05 ± 1.65e-06 | 4.53e-05 ± 2.57e-06 |
| Out-of-the | 32 | 1.92e-01 | 2.28e-01 ± 8.22e-02 | 2.32e-01 ± 4.92e-02 | 3.95e-01 ± 1.01e-01 |
| Money | 64 | 6.05e-02 | 8.39e-02 ± 4.89e-02 | 8.83e-02 ± 3.15e-02 | 1.83e-01 ± 5.91e-02 |
| | 128 | 2.20e-02 | 3.06e-02 ± 2.07e-02 | 3.33e-02 ± 1.42e-02 | 8.46e-02 ± 3.04e-02 |
| | 256 | 8.60e-03 | 1.25e-02 ± 1.05e-02 | 1.32e-02 ± 4.40e-03 | 3.75e-02 ± 1.22e-02 |
| | 512 | 3.47e-03 | 4.93e-03 ± 3.82e-03 | 4.07e-03 ± 8.52e-04 | 1.27e-02 ± 4.99e-03 |
| | 1024 | 1.46e-03 | 2.02e-03 ± 1.94e-03 | 1.62e-03 ± 3.36e-04 | 4.83e-03 ± 2.24e-03 |
| | 2048 | 6.19e-04 | 5.44e-04 ± 1.42e-05 | 5.93e-04 ± 4.38e-05 | 1.74e-03 ± 1.43e-03 |
| | 4096 | 2.58e-04 | 2.48e-04 ± 6.55e-06 | 2.57e-04 ± 2.43e-05 | 3.59e-04 ± 1.84e-04 |
| | 8192 | 1.17e-04 | 1.12e-04 ± 4.33e-06 | 1.10e-04 ± 1.18e-05 | 1.10e-04 ± 1.07e-05 |
| At-the | 32 | 3.25e-01 | 4.04e-01 ± 1.82e-01 | 4.12e-01 ± 1.12e-01 | 7.76e-01 ± 2.20e-01 |
| Money | 64 | 8.73e-02 | 1.38e-01 ± 1.08e-01 | 1.47e-01 ± 6.94e-02 | 3.56e-01 ± 1.27e-01 |
| | 128 | 2.77e-02 | 4.56e-02 ± 4.53e-02 | 5.11e-02 ± 3.10e-02 | 1.64e-01 ± 6.51e-02 |
| | 256 | 9.23e-03 | 1.77e-02 ± 2.26e-02 | 1.87e-02 ± 9.72e-03 | 7.20e-02 ± 2.65e-02 |
| | 512 | 3.25e-03 | 6.36e-03 ± 8.09e-03 | 4.36e-03 ± 1.79e-03 | 2.33e-02 ± 1.10e-02 |
| | 1024 | 1.23e-03 | 2.54e-03 ± 4.09e-03 | 1.51e-03 ± 6.94e-04 | 8.69e-03 ± 4.87e-03 |
| | 2048 | 5.50e-04 | 5.14e-04 ± 1.52e-05 | 5.36e-04 ± 2.67e-05 | 3.00e-03 ± 3.08e-03 |
| | 4096 | 2.40e-04 | 2.30e-04 ± 7.48e-06 | 2.33e-04 ± 1.03e-05 | 4.80e-04 ± 4.07e-04 |
| | 8192 | 1.02e-04 | 9.95e-05 ± 3.97e-06 | 9.50e-05 ± 6.18e-06 | 1.03e-04 ± 7.57e-06 |
| In-the | 32 | 1.40e-01 | 1.74e-01 ± 8.01e-02 | 1.76e-01 ± 4.77e-02 | 3.45e-01 ± 1.02e-01 |
| Money | 64 | 3.67e-02 | 5.85e-02 ± 4.74e-02 | 6.12e-02 ± 2.95e-02 | 1.61e-01 ± 6.00e-02 |
| | 128 | 1.06e-02 | 1.93e-02 ± 2.16e-02 | 2.07e-02 ± 1.40e-02 | 7.87e-02 ± 3.34e-02 |
| | 256 | 3.08e-03 | 6.98e-03 ± 1.03e-02 | 7.14e-03 ± 4.70e-03 | 3.47e-02 ± 1.33e-02 |
| | 512 | 8.68e-04 | 2.28e-03 ± 3.56e-03 | 1.45e-03 ± 1.06e-03 | 1.21e-02 ± 6.13e-03 |
| | 1024 | 2.61e-04 | 8.97e-04 ± 1.94e-03 | 4.25e-04 ± 4.24e-04 | 4.61e-03 ± 2.77e-03 |
| | 2048 | 1.01e-04 | 1.01e-04 ± 1.70e-06 | 1.10e-04 ± 1.02e-05 | 1.64e-03 ± 1.77e-03 |
| | 4096 | 4.04e-05 | 4.18e-05 ± 1.35e-06 | 4.36e-05 ± 4.54e-06 | 2.25e-04 ± 2.93e-04 |
| | 8192 | 1.77e-05 | 1.71e-05 ± 4.83e-07 | 1.72e-05 ± 8.41e-07 | 1.84e-05 ± 1.43e-06 |
| High | 32 | 2.08e+00 | 2.60e+00 ± 1.19e+00 | 2.64e+00 ± 7.19e-01 | 5.08e+00 ± 1.50e+00 |
| Volatility | 64 | 5.71e-01 | 9.00e-01 ± 7.05e-01 | 9.54e-01 ± 4.52e-01 | 2.38e+00 ± 8.77e-01 |
| | 128 | 1.81e-01 | 3.05e-01 ± 3.09e-01 | 3.36e-01 ± 2.07e-01 | 1.14e+00 ± 4.65e-01 |
| | 256 | 5.98e-02 | 1.17e-01 ± 1.51e-01 | 1.23e-01 ± 6.66e-02 | 5.02e-01 ± 1.86e-01 |
| | 512 | 2.05e-02 | 4.14e-02 ± 5.37e-02 | 2.86e-02 ± 1.37e-02 | 1.70e-01 ± 8.15e-02 |
| | 1024 | 7.56e-03 | 1.66e-02 ± 2.81e-02 | 9.77e-03 ± 5.41e-03 | 6.43e-02 ± 3.65e-02 |
| | 2048 | 3.15e-03 | 2.87e-03 ± 7.08e-05 | 3.14e-03 ± 2.15e-04 | 2.27e-02 ± 2.33e-02 |
| | 4096 | 1.29e-03 | 1.26e-03 ± 2.95e-05 | 1.31e-03 ± 1.14e-04 | 3.43e-03 ± 3.47e-03 |
| | 8192 | 5.66e-04 | 5.59e-04 ± 2.86e-05 | 5.38e-04 ± 7.12e-05 | 5.58e-04 ± 6.19e-05 |
| Low | 32 | 2.38e-02 | 2.92e-02 ± 1.24e-02 | 2.99e-02 ± 7.67e-03 | 5.41e-02 ± 1.46e-02 |
| Volatility | 64 | 6.57e-03 | 1.00e-02 ± 7.30e-03 | 1.07e-02 ± 4.73e-03 | 2.44e-02 ± 8.32e-03 |
| | 128 | 2.17e-03 | 3.34e-03 ± 2.98e-03 | 3.77e-03 ± 2.08e-03 | 1.10e-02 ± 4.18e-03 |
| | 256 | 7.56e-04 | 1.32e-03 ± 1.51e-03 | 1.41e-03 ± 6.36e-04 | 4.81e-03 ± 1.72e-03 |
| | 512 | 2.85e-04 | 4.95e-04 ± 5.44e-04 | 3.59e-04 ± 1.07e-04 | 1.51e-03 ± 6.71e-04 |
| | 1024 | 1.13e-04 | 1.99e-04 ± 2.66e-04 | 1.32e-04 ± 4.12e-05 | 5.60e-04 ± 2.97e-04 |
| | 2048 | 5.21e-05 | 4.87e-05 ± 1.38e-06 | 5.03e-05 ± 2.17e-06 | 1.92e-04 ± 1.85e-04 |
| | 4096 | 2.32e-05 | 2.22e-05 ± 7.64e-07 | 2.23e-05 ± 8.73e-07 | 3.53e-05 ± 2.18e-05 |
| | 8192 | 1.01e-05 | 9.66e-06 ± 3.41e-07 | 9.40e-06 ± 4.93e-07 | 1.00e-05 ± 6.33e-07 |

Table 10: The table compares the standard Sobol' sequence (Joe & Kuo) against three variants: our implementation (LLM), our implementation with the flash-lite model (LLM-lite), and our implementation starting from random initialization (LLM random init.) across all six Asian option scenarios. We report the Mean Squared Error (MSE).

| Scenario | $S_0$ | K | $\sigma$ | $S_{\text{true}}$ |
|---|---|---|---|---|
| Training Example | 50.00 | 45.00 | 0.3 | 7.06 |
| Out-of-the-Money | 50.00 | 60.00 | 0.3 | 1.02 |
| At-the-Money | 50.00 | 52.50 | 0.3 | 2.98 |
| In-the-Money | 50.00 | 40.00 | 0.3 | 11.02 |
| High Volatility | 50.00 | 52.50 | 0.6 | 6.43 |
| Low Volatility | 50.00 | 52.50 | 0.1 | 0.69 |

Table 11: Asian option scenarios used for testing. The Training Example was used in the evaluation routine of the evolutionary search. All options have $T = 1.0$ and $r = 0.05$.

| Option Type | Scenario | Initial Price ($S_0$) | Strike Price (K) | Volatility ($\sigma$) | Other Parameters |
|---|---|---|---|---|---|
| Asian | Training Example | 50.00 | 45.00 | 0.3 | — |
| | Out-of-the-Money | 50.00 | 60.00 | 0.3 | — |
| | At-the-Money | 50.00 | 52.50 | 0.3 | — |
| | In-the-Money | 50.00 | 40.00 | 0.3 | — |
| | High Volatility | 50.00 | 52.50 | 0.6 | — |
| | Low Volatility | 50.00 | 52.50 | 0.1 | — |
| Lookback | Base | 100.00 | — | 0.2 | — |
| | High Volatility | 100.00 | — | 0.4 | — |
| Barrier | Base | 100.00 | 100.00 | 0.2 | Barrier Level: 85.00 |
| | Close Barrier | 100.00 | 100.00 | 0.2 | Barrier Level: 95.00 |
| Basket (32D) | Low Correlation | 100.00 | 100.00 | 0.2 | $\rho$: 0.1 |
| | High Correlation | 100.00 | 100.00 | 0.2 | $\rho$: 0.8 |
| | Mixed Volatility | 100.00 | 100.00 | $U(0.15, 0.4)$ | $\rho$: 0.5 |
| | Out-of-the-Money | 100.00 | 110.00 | 0.2 | $\rho$: 0.1 |
| Bermudan | At-the-Money | 100.00 | 100.00 | 0.2 | Exercise Dates: 4 |
| | In-the-Money | 90.00 | 100.00 | 0.2 | Exercise Dates: 4 |

Table 12: Configuration Parameters for All Tested Option Scenarios. This table details the parameters for the 32-dimensional options used in the primary benchmark and generalizability tests. For all scenarios, the time to expiration (T) is 1.0 year and the risk-free interest rate (r) is 0.05. An em-dash (—) indicates a parameter is not applicable to that option type.

| Scenario | N | Squared Bias | | Variance | | MSE | | p-value |
|---|---|---|---|---|---|---|---|---|
| | | Sobol | LLM | Sobol | LLM | Sobol | LLM | |
| Training | 32 | 2.09e-05 | **5.59e-07** | **0.2484** | 0.2523 | **0.2484** | 0.2523 | 0.9757 |
| Example | 64 | **5.37e-06** | 9.58e-06 | **0.0642** | 0.0665 | **0.0642** | 0.0665 | 0.9980 |
| | 128 | 5.10e-06 | **4.96e-06** | **0.0183** | 0.0192 | **0.0183** | 0.0192 | 0.9999 |
| | 256 | 2.87e-07 | **1.43e-07** | **0.0056** | 0.0061 | **0.0056** | 0.0061 | 1.0000 |
| | 512 | 6.09e-08 | **3.51e-09** | 0.001646 | **0.001614** | 0.001646 | **0.001614** | 0.2841 |
| | 1024 | 7.89e-08 | **1.33e-08** | 0.000542 | **0.000524** | 0.000542 | **0.000524** | 0.0548 |
| | 2048 | 2.60e-08 | **2.30e-12** | 0.000233 | **0.000225** | 0.000233 | **0.000225** | **0.0199** |
| | 4096 | **9.11e-10** | 3.46e-09 | 9.876e-05 | **9.346e-05** | 9.876e-05 | **9.346e-05** | **0.0058** |
| | 8192 | 1.55e-09 | **1.93e-10** | 4.523e-05 | **4.104e-05** | 4.523e-05 | **4.104e-05** | **7.0e-06** |
| Out-of-the | 32 | 3.73e-05 | **6.41e-07** | **0.1920** | 0.1979 | **0.1920** | 0.1979 | 0.9757 |
| Money | 64 | **1.48e-06** | 9.78e-09 | **0.0605** | 0.0631 | **0.0605** | 0.0631 | 0.9980 |
| | 128 | **3.74e-07** | 1.66e-06 | **0.0220** | 0.0231 | **0.0220** | 0.0231 | 0.9999 |
| | 256 | 3.07e-07 | **2.60e-07** | **0.0086** | 0.0089 | **0.0086** | 0.0089 | 1.0000 |
| | 512 | 4.47e-08 | **4.05e-11** | 0.003469 | **0.003311** | 0.003469 | **0.003311** | **0.0208** |
| | 1024 | **1.29e-08** | 4.45e-08 | 0.001457 | **0.001313** | 0.001457 | **0.001313** | **3.5e-08** |
| | 2048 | **2.92e-11** | 3.36e-08 | 0.000619 | **0.000535** | 0.000619 | **0.000535** | **1.5e-14** |
| | 4096 | 5.09e-08 | **1.17e-09** | 0.000258 | **0.000243** | 0.000258 | **0.000243** | **4.5e-04** |
| | 8192 | 3.71e-09 | **3.24e-09** | 0.000117 | **0.000107** | 0.000117 | **0.000107** | **1.0e-06** |
| At-the | 32 | 5.90e-05 | **1.35e-09** | **0.3246** | 0.3346 | **0.3246** | 0.3346 | 0.9757 |
| Money | 64 | 1.35e-05 | **1.17e-05** | **0.0873** | 0.0908 | **0.0873** | 0.0908 | 0.9980 |
| | 128 | 9.72e-06 | **1.45e-06** | **0.0277** | 0.0282 | **0.0277** | 0.0282 | 0.9999 |
| | 256 | 1.12e-06 | **2.06e-08** | **0.0092** | 0.0098 | **0.0092** | 0.0098 | 1.0000 |
| | 512 | 1.83e-08 | **5.57e-10** | 0.003248 | **0.003106** | 0.003248 | **0.003106** | **0.0208** |
| | 1024 | 3.60e-08 | **4.58e-09** | 0.001228 | **0.001168** | 0.001228 | **0.001168** | **0.0304** |
| | 2048 | **7.85e-11** | 1.82e-08 | 0.000550 | **0.000521** | 0.000550 | **0.000521** | **0.0131** |
| | 4096 | **3.16e-10** | 8.98e-10 | 0.000240 | **0.000226** | 0.000240 | **0.000226** | **0.0019** |
| | 8192 | 7.47e-09 | **3.31e-09** | 0.000102 | **9.522e-05** | 0.000102 | **9.523e-05** | **0.0100** |
| In-the | 32 | 1.93e-05 | **3.11e-09** | **0.1397** | 0.1428 | **0.1398** | 0.1428 | 0.9757 |
| Money | 64 | **2.30e-06** | 2.86e-06 | **0.0367** | 0.0382 | **0.0367** | 0.0382 | 0.9980 |
| | 128 | **1.67e-06** | 1.74e-06 | **0.0106** | 0.0111 | **0.0106** | 0.0111 | 0.9999 |
| | 256 | **1.08e-09** | 3.54e-08 | **0.003083** | 0.003373 | **0.003083** | 0.003373 | 1.0000 |
| | 512 | **3.07e-10** | 4.26e-08 | 0.000868 | **0.000859** | 0.000868 | **0.000859** | 0.2841 |
| | 1024 | 1.53e-09 | **6.44e-10** | 0.000261 | **0.000260** | 0.000261 | **0.000260** | 0.2581 |
| | 2048 | **3.54e-10** | 1.28e-08 | 0.000101 | **0.000100** | 0.000101 | **0.000100** | 0.3878 |
| | 4096 | **2.15e-11** | 3.38e-09 | 4.037e-05 | **3.949e-05** | 4.037e-05 | **3.949e-05** | 0.2180 |
| | 8192 | **2.77e-10** | 3.70e-09 | 1.766e-05 | **1.682e-05** | 1.766e-05 | **1.683e-05** | **0.0047** |
| High | 32 | 0.000333 | **2.83e-07** | **2.0813** | 2.1493 | **2.0816** | 2.1493 | 0.9757 |
| Volatility | 64 | 5.87e-05 | **4.32e-05** | **0.5713** | 0.5958 | **0.5714** | 0.5958 | 0.9980 |
| | 128 | 4.41e-05 | **5.25e-06** | **0.1810** | 0.1877 | **0.1811** | 0.1877 | 0.9999 |
| | 256 | **1.78e-06** | 1.90e-06 | **0.0598** | 0.0643 | **0.0598** | 0.0643 | 1.0000 |
| | 512 | **2.34e-09** | 1.49e-08 | 0.0205 | **0.0196** | 0.0205 | **0.0196** | **0.0073** |
| | 1024 | **1.21e-07** | 2.19e-07 | 0.007565 | **0.007060** | 0.007565 | **0.007060** | **3.4e-04** |
| | 2048 | **6.34e-08** | 1.22e-07 | 0.003145 | **0.002867** | 0.003145 | **0.002867** | **1.8e-06** |
| | 4096 | **2.93e-10** | 6.09e-09 | 0.001293 | **0.001233** | 0.001293 | **0.001233** | **0.0022** |
| | 8192 | **3.59e-08** | 4.35e-08 | 0.000566 | **0.000532** | 0.000566 | **0.000532** | **0.0156** |
| Low | 32 | 5.75e-06 | **1.26e-08** | **0.0238** | 0.0246 | **0.0238** | 0.0246 | 0.9757 |
| Volatility | 64 | 1.28e-06 | **6.48e-07** | **0.0066** | 0.0068 | **0.0066** | 0.0068 | 0.9980 |
| | 128 | 7.71e-07 | **2.71e-08** | **0.002173** | 0.002205 | **0.002174** | 0.002205 | 0.9999 |
| | 256 | 1.08e-07 | **1.12e-08** | **0.000756** | 0.000803 | **0.000756** | 0.000803 | 1.0000 |
| | 512 | 6.51e-09 | **1.55e-09** | 0.000285 | **0.000275** | 0.000285 | **0.000275** | **0.0208** |
| | 1024 | 5.18e-09 | **1.20e-09** | 0.000113 | **0.000109** | 0.000113 | **0.000109** | 0.0717 |
| | 2048 | **2.66e-10** | 2.17e-09 | 5.214e-05 | **4.961e-05** | 5.214e-05 | **4.961e-05** | **0.0056** |
| | 4096 | **1.26e-10** | 1.39e-10 | 2.323e-05 | **2.196e-05** | 2.323e-05 | **2.196e-05** | **5.7e-04** |
| | 8192 | 3.43e-10 | **3.09e-10** | 1.008e-05 | **9.300e-06** | 1.008e-05 | **9.300e-06** | **1.3e-04** |

Table 13: Integration results for the Asian option across all six scenarios. The table compares the standard Sobol' sequence (Joe & Kuo) against the direction numbers discovered by LLM evolutionary search. We report the Mean Squared Error (MSE), its constituent parts (Squared Bias and Variance), and the FDR-corrected p-value from a one-sided Wilcoxon signed-rank test. P-values below 0.05 and the best methods are **bolded**.

| Scenario | N | Squared Bias | | Variance | | MSE | | p-value |
|---|---|---|---|---|---|---|---|---|
| | | Sobol | LLM | Sobol | LLM | Sobol | LLM | |
| Base | 32 | **3.90e-06** | 8.56e-05 | **1.2138** | 1.2443 | **1.2138** | 1.2444 | 0.9968 |
| | 64 | **4.16e-06** | 3.01e-05 | **0.4725** | 0.4839 | **0.4725** | 0.4840 | 1.0000 |
| | 128 | 3.14e-06 | **2.35e-07** | **0.1272** | 0.1352 | **0.1272** | 0.1352 | 1.0000 |
| | 256 | **1.61e-06** | 4.90e-06 | **0.0352** | 0.0381 | **0.0352** | 0.0381 | 1.0000 |
| | 512 | **8.86e-07** | 1.10e-06 | 0.0121 | **0.0120** | 0.0121 | **0.0120** | 0.4675 |
| | 1024 | **9.64e-07** | 1.43e-06 | 0.005021 | **0.004905** | 0.005022 | **0.004907** | **0.0342** |
| | 2048 | 6.88e-08 | **1.29e-07** | 0.001834 | **0.001744** | 0.001834 | **0.001744** | **4.1e-04** |
| | 4096 | 1.32e-07 | **5.59e-08** | 0.000807 | **0.000768** | 0.000807 | **0.000768** | **2.4e-04** |
| | 8192 | 5.21e-08 | **3.61e-08** | 0.000407 | **0.000377** | 0.000407 | **0.000377** | **9.8e-09** |
| High | 32 | **2.13e-04** | 4.44e-04 | **8.5591** | 8.7869 | **8.5593** | 8.7873 | 0.9968 |
| Volatility | 64 | **6.04e-08** | 1.16e-04 | **3.3964** | 3.5134 | **3.3964** | 3.5136 | 1.0000 |
| | 128 | 2.87e-05 | **2.57e-06** | **0.8854** | 0.9982 | **0.8854** | 0.9982 | 1.0000 |
| | 256 | **3.14e-06** | 1.46e-05 | **0.2403** | 0.2663 | **0.2403** | 0.2663 | 1.0000 |
| | 512 | 7.96e-07 | **1.11e-07** | 0.0809 | **0.0792** | 0.0809 | **0.0792** | 0.3836 |
| | 1024 | **2.57e-06** | 7.04e-06 | 0.0325 | **0.0312** | 0.0325 | **0.0312** | **6.2e-04** |
| | 2048 | 3.37e-07 | **4.92e-08** | 0.012003 | **0.011234** | 0.012003 | **0.011234** | **2.3e-06** |
| | 4096 | 1.47e-07 | **2.32e-08** | 0.004862 | **0.004558** | 0.004862 | **0.004558** | **2.7e-06** |
| | 8192 | **2.12e-10** | 6.25e-09 | 0.002335 | **0.002057** | 0.002335 | **0.002057** | **9.8e-17** |

Table 14: Integration results for the Lookback Option across two scenarios. We report FDR-corrected P-values from a one-sided Wilcoxon signed-rank test. P-values below 0.05 and the best methods are **bolded**.

| Scenario | N | Squared Bias | | Variance | | MSE | | p-value |
|---|---|---|---|---|---|---|---|---|
| | | Sobol | LLM | Sobol | LLM | Sobol | LLM | |
| Base | 32 | 1.13e-04 | **2.87e-05** | **2.2540** | 2.3339 | **2.2541** | 2.3339 | 0.9968 |
| | 64 | 2.17e-05 | **4.56e-06** | **0.8386** | 0.8894 | **0.8386** | 0.8894 | 1.0000 |
| | 128 | **2.32e-05** | 2.37e-05 | **0.2533** | 0.3101 | **0.2534** | 0.3101 | 1.0000 |
| | 256 | **1.97e-06** | 7.40e-06 | **0.0707** | 0.0834 | **0.0707** | 0.0834 | 1.0000 |
| | 512 | **5.72e-07** | 4.84e-06 | 0.0245 | **0.0241** | 0.0245 | **0.0241** | 0.4675 |
| | 1024 | 7.63e-07 | **7.40e-07** | 0.009948 | **0.009819** | 0.009948 | **0.009819** | 0.3313 |
| | 2048 | **4.03e-07** | 1.12e-06 | 0.004277 | **0.004230** | 0.004277 | **0.004231** | 0.3025 |
| | 4096 | **8.61e-08** | 1.00e-07 | 0.002019 | **0.001992** | 0.002019 | **0.001992** | 0.3863 |
| | 8192 | **1.82e-08** | 5.70e-08 | 0.000985 | **0.000955** | 0.000985 | **0.000955** | 0.0859 |
| Close | 32 | 9.55e-04 | **1.27e-04** | **3.2150** | 3.2754 | **3.2160** | 3.2755 | 0.9968 |
| Barrier | 64 | 6.55e-05 | **1.44e-05** | **1.1229** | 1.2054 | **1.1229** | 1.2054 | 1.0000 |
| | 128 | 5.52e-05 | **1.14e-05** | **0.4541** | 0.5183 | **0.4542** | 0.5183 | 1.0000 |
| | 256 | 3.62e-05 | **8.39e-07** | **0.1733** | 0.1775 | **0.1733** | 0.1775 | 1.0000 |
| | 512 | 3.76e-05 | **7.16e-07** | 0.0723 | **0.0690** | 0.0723 | **0.0690** | 0.2364 |
| | 1024 | 2.27e-05 | **2.62e-06** | 0.0328 | **0.0326** | 0.0328 | **0.0326** | 0.4459 |
| | 2048 | 1.52e-06 | **1.13e-09** | **0.01594** | 0.01648 | **0.01594** | 0.01648 | 0.9126 |
| | 4096 | **3.83e-07** | 4.25e-07 | 0.007027 | **0.006727** | 0.007028 | **0.006728** | 0.0574 |
| | 8192 | **3.48e-09** | 3.79e-07 | 0.003379 | **0.003266** | 0.003379 | **0.003267** | 0.1613 |

Table 15: Integration results for the Barrier Option across two scenarios. We report FDR-corrected p-values from a one-sided Wilcoxon signed-rank test. P-values below 0.05 and the best methods are **bolded**.

| Scenario | N | Squared Bias | | Variance | | MSE | | p-value |
|---|---|---|---|---|---|---|---|---|
| | | Sobol | LLM | Sobol | LLM | Sobol | LLM | |
| Low | 32 | 3.82e-06 | **5.13e-07** | **0.115** | 0.118 | **0.115** | 0.118 | 0.997 |
| Correlation | 64 | **6.60e-07** | 1.99e-06 | **0.0342** | 0.0358 | **0.0342** | 0.0358 | 1.00 |
| | 128 | **1.18e-06** | 1.38e-06 | **0.0109** | 0.0126 | **0.0109** | 0.0126 | 1.00 |
| | 256 | 5.16e-08 | **1.38e-08** | **0.00360** | 0.00390 | **0.00360** | 0.00390 | 1.00 |
| | 512 | 3.04e-07 | **1.15e-08** | 0.00126 | **0.00123** | 0.00126 | **0.00123** | 0.238 |
| | 1024 | 2.00e-07 | **1.79e-08** | 5.04e-04 | **4.73e-04** | 5.04e-04 | **4.73e-04** | **0.0051** |
| | 2048 | 5.00e-08 | **6.46e-11** | 2.29e-04 | **2.14e-04** | 2.29e-04 | **2.14e-04** | **3.8e-04** |
| | 4096 | **1.12e-10** | 1.32e-09 | 9.89e-05 | **9.04e-05** | 9.89e-05 | **9.04e-05** | **1.7e-04** |
| | 8192 | 1.39e-09 | **1.72e-10** | 4.74e-05 | **4.16e-05** | 4.74e-05 | **4.16e-05** | **1.9e-11** |
| High | 32 | **9.26e-05** | 9.80e-05 | **0.248** | 0.256 | **0.248** | 0.256 | 0.997 |
| Correlation | 64 | **7.77e-06** | 1.10e-05 | 0.0602 | **0.0570** | 0.0602 | **0.0571** | **2.6e-05** |
| | 128 | 6.94e-06 | **2.65e-06** | **0.0163** | 0.0168 | **0.0163** | 0.0168 | 1.00 |
| | 256 | **7.96e-07** | 8.54e-07 | **0.00491** | 0.00494 | **0.00491** | 0.00495 | 1.00 |
| | 512 | **6.53e-08** | 8.95e-08 | 0.00153 | **0.00152** | 0.00153 | **0.00152** | 0.468 |
| | 1024 | **3.80e-10** | 1.16e-08 | **4.63e-04** | 4.47e-04 | **4.63e-04** | 4.47e-04 | 0.0784 |
| | 2048 | 3.45e-09 | **1.57e-09** | **1.75e-04** | 1.78e-04 | **1.75e-04** | 1.78e-04 | 0.742 |
| | 4096 | **8.68e-10** | 7.39e-09 | 7.13e-05 | **6.61e-05** | 7.13e-05 | **6.61e-05** | **8.2e-04** |
| | 8192 | 6.90e-09 | **1.17e-10** | 2.60e-05 | **2.45e-05** | 2.60e-05 | **2.45e-05** | **0.00530** |
| Mixed | 32 | 2.19e-04 | **9.65e-05** | **0.775** | 0.799 | **0.776** | 0.799 | 0.997 |
| Volatility | 64 | **9.78e-06** | 1.28e-05 | 0.210 | **0.207** | 0.210 | **0.207** | 0.540 |
| | 128 | 1.95e-05 | **2.19e-06** | **0.0647** | 0.0702 | **0.0647** | 0.0702 | 1.00 |
| | 256 | 4.42e-06 | **1.52e-06** | **0.0211** | 0.0216 | **0.0211** | 0.0216 | 1.00 |
| | 512 | 6.51e-07 | **6.07e-07** | 0.00620 | **0.00611** | 0.00620 | **0.00611** | 0.640 |
| | 1024 | 1.17e-08 | **5.20e-09** | 0.00209 | **0.00203** | 0.00209 | **0.00203** | 0.0784 |
| | 2048 | 1.09e-08 | **4.65e-11** | 9.47e-04 | **9.32e-04** | 9.47e-04 | **9.32e-04** | 0.0672 |
| | 4096 | **3.40e-09** | 2.79e-08 | 4.47e-04 | **3.93e-04** | 4.47e-04 | **3.93e-04** | **1.5e-10** |
| | 8192 | 1.80e-08 | **2.19e-12** | 1.56e-04 | **1.43e-04** | 1.56e-04 | **1.43e-04** | **1.5e-05** |
| Out-of-the | 32 | 2.43e-05 | **2.29e-09** | **0.117** | 0.121 | **0.117** | 0.121 | 0.997 |
| Money | 64 | 2.81e-06 | **4.90e-09** | **0.0367** | 0.0387 | **0.0367** | 0.0387 | 1.00 |
| | 128 | 2.55e-06 | **2.01e-07** | **0.0127** | 0.0143 | **0.0127** | 0.0143 | 1.00 |
| | 256 | 5.97e-07 | **4.26e-10** | **0.00437** | 0.00463 | **0.00437** | 0.00463 | 1.00 |
| | 512 | 2.65e-07 | **2.67e-08** | 0.00160 | **0.00155** | 0.00160 | **0.00155** | 0.236 |
| | 1024 | 3.70e-08 | **2.12e-09** | 6.52e-04 | **6.23e-04** | 6.52e-04 | **6.23e-04** | 0.0651 |
| | 2048 | 5.68e-09 | **3.00e-11** | 2.92e-04 | **2.81e-04** | 2.92e-04 | **2.81e-04** | 0.0856 |
| | 4096 | 1.01e-08 | **5.67e-09** | 1.25e-04 | **1.20e-04** | 1.25e-04 | **1.20e-04** | **0.0027** |
| | 8192 | **1.58e-11** | 1.66e-09 | 5.95e-05 | **5.39e-05** | 5.95e-05 | **5.39e-05** | **2.5e-06** |

Table 16: Integration results for the 32-dimensional Basket Option across four scenarios. We report FDR-corrected p-values from a one-sided Wilcoxon signed-rank test. P-values below 0.05 and the best methods are **bolded**.

| Scenario | N | Squared Bias | | Variance | | MSE | | p-value |
|---|---|---|---|---|---|---|---|---|
| | | Sobol | LLM | Sobol | LLM | Sobol | LLM | |
| At-the | 32 | **0.817** | 0.827 | **0.574** | 0.583 | **1.39** | 1.41 | 0.997 |
| Money | 64 | **0.254** | 0.263 | 0.240 | **0.237** | **0.495** | 0.500 | 1.00 |
| | 128 | **0.0698** | 0.0725 | **0.0941** | 0.0951 | **0.164** | 0.168 | 1.00 |
| | 256 | **0.0182** | 0.0192 | **0.0441** | 0.0445 | **0.0624** | 0.0637 | 1.00 |
| | 512 | 0.00440 | **0.00410** | **0.0218** | 0.0221 | **0.0262** | 0.0262 | 0.468 |
| | 1024 | 9.87e-04 | **9.33e-04** | 0.0118 | **0.0112** | 0.0128 | **0.0122** | **0.0115** |
| | 2048 | 2.77e-04 | **2.46e-04** | 0.00596 | **0.00571** | 0.00623 | **0.00596** | **0.0018** |
| | 4096 | 5.23e-05 | **4.25e-05** | 0.00270 | **0.00268** | 0.00275 | **0.00272** | 0.386 |
| | 8192 | 1.10e-05 | **8.45e-06** | 0.00132 | **0.00127** | 0.00134 | **0.00128** | **0.0457** |
| In-the | 32 | **0.889** | 0.898 | 0.563 | **0.555** | **1.45** | 1.45 | 0.997 |
| Money | 64 | **0.223** | 0.230 | 0.251 | **0.251** | **0.473** | 0.481 | 1.00 |
| | 128 | **0.0478** | 0.0489 | 0.118 | **0.117** | 0.166 | **0.166** | 1.00 |
| | 256 | **0.00920** | 0.00930 | 0.0564 | **0.0563** | **0.0656** | **0.0656** | 1.00 |
| | 512 | 0.00200 | **0.00190** | 0.0284 | **0.0278** | 0.0304 | **0.0297** | 0.468 |
| | 1024 | **3.75e-04** | 4.09e-04 | 0.0145 | **0.0135** | 0.0149 | **0.0139** | **0.0142** |
| | 2048 | **6.75e-05** | 7.12e-05 | 0.00735 | **0.00658** | 0.00742 | **0.00666** | **1.2e-05** |
| | 4096 | 1.17e-05 | **9.36e-06** | 0.00337 | **0.00313** | 0.00338 | **0.00314** | **7.4e-04** |
| | 8192 | **3.56e-06** | 3.86e-06 | 0.00166 | **0.00152** | 0.00166 | **0.00153** | **1.6e-04** |

Table 17: Integration results for the Bermudan Option across two scenarios. We report FDR-corrected p-values from a one-sided Wilcoxon signed-rank test. P-values below 0.05 and the best methods are **bolded**.

