# OpenReview forum: "LLM-Guided Evolutionary Program Synthesis for Quasi-Monte Carlo Design"
_ICLR.cc/2026/Conference — ICLR 2026 Poster_

### Official Review · Reviewer_jdvu · 2025-10-16

**Soundness:** 2
**Presentation:** 2
**Contribution:** 2
**Rating:** 2
**Confidence:** 3

**Summary:**

This paper proposes a new methodology to find low-discrepancy point sets, by using an evolutionary algorithm where the mutation phase is done by an LLM. Two settings are inspected: in the first setting, the goal is to minimize the star discrepancy of a point set in 2D and 3D. Here the generation space is the space of Python programs. In the second setting, the goal is to minimize the integration error for a 32D option-pricing task. Here the generation space is the set of parameters (so-called direction numbers) for Sobol digital nets. In both cases, the solution found by the proposed method outperforms baselines.

**Strengths:**

Guiding program synthesis through LLMs is a thriving avenue of research, which has shown progress in various fields. I appreciate that this paper brings this idea to the problem of low-discrepancy point set generation. The experiments show an improvement over the best known star-discrepancy sets in 2D and 3D, and improve over Joe and Kuo direction numbers in the Asian option pricing experiment.

**Weaknesses:**

My main concern is the lack of scientific evaluation, especially given that the proposed method is highly resource-intensive, involving thousands of LLM calls, each requiring computing the fitness of the point set. Such evaluation is critical to assess the proposed approach. Examples of lacking scientific evaluation include:
- comparing with simpler methods for exploring the space of low-discrepancy sequences. I discuss this point below for each experimental setting.
- reporting the improvement in terms of fitness during the run of the evolutionary algorithm.
- comparing with simple (single- or multi-turn) LLM prompting without population-based evolution.

For the star-discrepancy experiment, a natural baseline is local optimization from an existing high-quality construction. Since the two-phase prompting strategy already instructs the LLM to “use scipy optimization routines such as scipy.optimize.minimize,” it would be appropriate to compare against standard optimizers (e.g., L-BFGS-B, SLSQP) initialized from Clément et al. or other RQMC sequences, using a comparable computational budget (e.g., 2000 randomizations).

For the Asian call experiment, the LLM modifies direction numbers in dimensions 4–6 relative to the Joe and Kuo numbers. Since the search space over these dimensions has size 2^12, comparable to the 2000 LLM calls reported, it is unclear if the proposed algorithm provides an advantage over random search. Including such comparison (and comparing to exhaustive search to get an indication of how far away the solution found by the LLM is from the optimum) would strengthen the analysis. Another baseline is the LatNetBuilder software, which is specifically designed for identifying good low-discrepancy point sets, including Sobol sequences. An interesting approach would be to rank direction numbers in terms of some fitness measure (e.g., star-discrepancy t-value) using the software, then to evaluate them in this order on the Asian call experiment, and to compare the performance with the LLM-guided evolutionary approach, as a function of the number of tested direction numbers.

**Questions:**

How many different direction numbers were evaluated when running the evolutionary algorithm on the option pricing problem? Is it 2000 as reported in Appendix B.3?

---

> ### Author Response · Authors · 2025-11-26
>
> Thank you for your constructive comments highlighting the need for rigorous evaluation given the computational cost.
>
> 1. comparing with simpler methods for exploring the space of low-discrepancy sequences. I discuss this point below for each experimental setting.
>
> To address this concern for point sets, we have included baselines that run SLSQP directly on existing high-quality constructions in Appendix B.5. We apply SLSQP (same objective and similar iteration budgets) starting from (i) Sobol’, (ii) Fibonacci lattices, and (iii) the best Phase I sets. These SLSQP-only baselines substantially improve over their seeds, but the LLM-evolved Phase II programs still achieve lower star discrepancy in 22 of the 24 tested values of N, with reductions up to 15% relative to the best SLSQP-only baseline. Clément et al. are by design locally optimal and we do not include them in the analysis.
>
> To address this concern for Sobol' direction numbers, we compare our results to those obtained by  LatNetBuilder. We run LatNetBuilder’s recommended random-CBC search for 2000 iterations (matching computational budget) with a projection-dependent t-value criterion. We then compare rQMC MSE of LatNetBuilder’s digital nets against our LLM-discovered Sobol’ sequence for the same six 32D Asian-style options. For all six payoffs, our sequence outperforms or matches LatNetBuilder.
>
> 2. reporting the improvement in terms of fitness during the run of the evolutionary algorithm.
>
> We agree this is important and have added this to Appendix B.4, which includes Figure 4, plotting best-so-far star discrepancy vs. iteration for several N (30–100). Each curve shows a clear downward trajectory as the evolutionary run progresses. The majority of the improvement happens early on, but occasional drops in star discrepancy are observed in later iterations.
>
> 3. comparing with simple (single- or multi-turn) LLM prompting without population-based evolution.
>
> We now include experiments that explicitly compare the evolutionary loop to simpler prompting strategies in Appendix B.6. For the 2D star-discrepancy task, we consider single-turn and multi-turn prompting where the LLM is asked to directly output code for point sets, without an evolutionary population. We run these from multiple seeds and compare final discrepancies. As summarized in Table 4 and Figure 5, the best purely prompted runs can get close to the best evolutionary runs on some seeds, but they are less robust: the evolutionary pipeline produces tight clusters of strong solutions across seeds, whereas multi-turn prompting alone is far more variable. We find single-turn prompting performs quite poorly.
>
> 4. “Number of different direction numbers evaluated; is it 2000?”
>
> Yes. We have clarified this now  in Appendix A.4. We cap the number of distinct Sobol programs (each corresponding to a concrete set of direction numbers) at K ≤ 2000 per evolutionary run, which matches the “2000 LLM calls” mentioned in Appendix B.3. This ensures that comparisons against SLSQP or LatNetBuilder-based baselines can be fairly budgeted.
>
> 5. Overall scientific evaluation given computational cost.
>
> We share your concern that high computational cost demands careful evaluation. While point set generation is impractical at high N and d due to the O(N^(d/2 + 1)) computational complexity of computing the exact star discrepancy, we stress that this is a one-time cost. Furthermore, once optimal Sobol' direction numbers have been found for a problem of interest, they can be applied for that problem at that dimensionality for any number of points. In practice, the majority of compute is spent either running the LLM proposed program, which we cap with a wall clock time of ten minutes, or evaluating the fitness function. Our LLM calls are relatively small, at a maximum of 8192 tokens per iteration with only 2000 such iterations being required.

---

> > ### Comment · Reviewer_jdvu · 2025-11-27
> >
> > Thanks for the rebuttal. I appreciate that the authors added additional experimental results, comparisons, and ablation studies, which significantly improves the soundness of the paper. For the star-discrepancy experiment, the gain with respect to the Fibonacci/local search is minimal. For the Asian call experiment, as already discussed in the initial review, the number of tested direction numbers (2000) is around half of the total size of the searched space given which direction numbers are modified (2^12 = 4096). While I acknowledge that the LLM automatically chooses which direction numbers to modify, this still limits the significance of the results in my opinion. Given all of this, I increase my score from 2 to 4.

---

### Official Review · Reviewer_W9ms · 2025-10-25

**Soundness:** 2
**Presentation:** 2
**Contribution:** 2
**Rating:** 4
**Confidence:** 2

**Summary:**

This paper introduces an LLM-guided evolutionary framework to automate the discovery of quasi-Monte Carlo (QMC) constructions. The authors treat the design of low-discrepancy point sets and Sobol’ sequence direction numbers as a program synthesis problem. Within the proposed OpenEvolve framework, an LLM iteratively mutates Python programs that generate candidate point sets or Sobol’ parameters, guided by fitness scores (e.g., inverse star discrepancy or rQMC mean squared error).

**Strengths:**

1. Casting QMC design as program synthesis is conceptually elegant and connects symbolic LLM reasoning with continuous numerical optimization.

2. The paper carefully combines constructive heuristics with iterative optimization, and the experimental evaluation uses statistically robust paired tests

**Weaknesses:**

1. The “LLM-guided” aspect is somewhat opaque. How much improvement stems from the LLM’s structured code editing versus brute-force evolutionary search or the built-in optimizer (SLSQP)?

2. This problem setting seems too narrow, and i am not an expert in this domain so i do not understand the significance of this improvement.

3. Are these compared baselines the best baseline in this field? Is there some other learning-based baselines? '


4. In my view, it seems the authors only carefully design a prompting procedure to solve a specfic problem. Not sure these design strategies can be used for other problems.

5. The scale N/d is kind of limited, and we do not know whether this method can scale. In my understanding, LLM can be quite hard to scale for high-dimensional numbers.

**Questions:**

See Weakness.

---

> ### Author Response · Authors · 2025-11-26
>
> We thank the reviewer for their constructive comments and for highlighting where the paper felt opaque.
>
> 1. The “LLM-guided” aspect is somewhat opaque. How much improvement stems from the LLM’s structured code editing versus brute-force evolutionary search or the built-in optimizer (SLSQP)?
>
> We address this now in Appendix B.5. We compare our method against applying SLSQP directly to Sobol, Fibonacci, and Phase-I point sets without the LLM's evolutionary logic (Table 3).  The LLM-evolved programs achieved strictly lower star discrepancy than the SLSQP baselines in almost every case. We believe this demonstrates that the improvement is not solely due to the built-in optimizer. The LLM discovers specific initialization strategies and restart heuristics that allow the classical optimizer to escape local minima better than standard applications of SLSQP.
>
> 2. This problem setting seems too narrow, and i am not an expert in this domain so i do not understand the significance of this improvement.
>
> While the problems we tackle are specialized, they are central to QMC: constructing low-discrepancy finite sets and Sobol' sequences underpins QMC methods used in financial engineering, computer graphics, and uncertainty quantification. For finite sets, we match all known 2D optima for N ≤ 21 and 3D optima for N ≤ 8 from Clément et al. (2024), and we report improved 3D benchmarks for N>8 where the optima are unknown. For larger 2D N (140–1020), we consistently beat classical sequences and the recent learning-based MPMC method across all tested N, sometimes by 15–20% in discrepancy. We believe these new point sets as well as using LLM to discover improved Sobol' direction numbers are of potential interest to the QMC community.
>
> 3. Are these compared baselines the best baseline in this field? Is there some other learning-based baselines?
>
> We compare our method to classical sequences, such as Sobol', Halton, Hammersley, Fibonacci, and rank-1 lattices as well as a recent learning-based method: MPMC (message-passing Monte Carlo) in 2D, which we use as the primary machine-learning baseline for finite sets. In most cases, our method matches the star discrepancy of provably optimal point sets. For Sobol' direction numbers, we compare against LatNetBuilder digital nets in Appendix C.1, representing state-of-the-art constructive search for digital sequences. Our evolved nets achieve lower MSE on option-pricing tasks.
>
> 4. In my view, it seems the authors only carefully design a prompting procedure to solve a specfic problem. Not sure these design strategies can be used for other problems.
>
> Our prompting is tailored to the type of object (point set vs. Sobol parameters), but the framework itself is generic: We use the same evolutionary loop across all tasks, changing only (i) the fitness function and (ii) a short natural-language description of the goal (e.g., “minimize star discrepancy” vs. “minimize rQMC MSE for a 32-dimensional Asian option”). The method is applied to two quite different problems (finite 2D/3D sets and 32D digital nets), without retraining the LLM and could be applied to new problems with only changes needed to the LLM prompt and fitness function, which can be quickly designed by a domain expert.
>
> 5. The scale N/d is kind of limited, and we do not know whether this method can scale. In my understanding, LLM can be quite hard to scale for high-dimensional numbers.
>
> Star discrepancy in high dimension is computationally intractable, which is why we restrict finite-set experiments to 2D and 3D. For higher dimensions, we switch to Sobol' direction numbers in d=32 with rQMC MSE as the fitness, which is scalable and directly relevant to practical integration tasks. We agree that scaling to much higher d (e.g., >100) or more complex integrands is an important direction. We can discuss this limitation and the potential for combining our approach with surrogate fitness or dimensionality-reduction techniques.

---

### Official Review · Reviewer_qCAy · 2025-10-31

**Soundness:** 4
**Presentation:** 4
**Contribution:** 4
**Rating:** 8
**Confidence:** 4

**Summary:**

The paper applies LLM to perform numerical integration with an evolutionary computation angle to improve the integration results.
Overall a nice paper.

**Strengths:**

Well written paper,
The application of LLM with an evolutionary loop to generate populations is interesting
The method has an evolving loop of introducing mutations to provide better solutions.

**Weaknesses:**

It appears that the approach provides better solution with N becomes larger and larger. However, the improvement is marginal
Over SOBOL and clement et. Al in the sense of MSE. While this marginal improvement is great, the computational complexity of
Running an evolutionary loop is something unknown right now.

**Questions:**

- P values to the order of 10^-14 indicate a  population with very little variance, is there any reason for such lack of variance?

- I think, this is a very interesting application, However, one needs to understand and study the computational complexity of getting such solutions. Moreover, I get that there are about 3 applications studied in this paper, I would like to know that generality of this approach, do you have to fine tune the LLM in some way.

- However, about the mutation, does the LLM involve some kind of RL based training to generate better mutations.

---

> ### Author Response · Authors · 2025-11-26
>
> We thank the reviewer for the positive assessment and for their thoughtful questions.
>
> 1. P values to the order of 10^-14 indicate a population with very little variance, is there any reason for such lack of variance?
>
> We use a paired one-sided Wilcoxon signed-rank test across 10,000 random scrambles and shifts per method and scenario. Because we pair randomizations (same random seed for each method), the variance of the difference in squared error across randomizations is small. When one method consistently yields lower error across almost all randomizations, the Wilcoxon statistic accumulates evidence rapidly, leading to very small p-values (often far below 10⁻⁶).
>
> 2. I think, this is a very interesting application, However, one needs to understand and study the computational complexity of getting such solutions. Moreover, I get that there are about 3 applications studied in this paper, I would like to know that generality of this approach, do you have to fine tune the LLM in some way.
>
> We have added a section (Appendix A.4) that provides a detailed complexity analysis. For Sobol' direction numbers, a single fitness evaluation requires R=1000 randomized QMC estimators with N=8192 and d=32, so each fitness evaluation processes ≈8.2×10⁶ 32-dimensional points. With at most K ≤ 2000 candidate programs per run, a full evolutionary run processes ≈1.6×10¹⁰ samples. As you point out, this is computationally heavy, but crucially it is offline and one-shot: once good parameters are found, using them has the same cost as using standard Sobol' sequences.
>
> We deliberately do not fine-tune the LLM or do any RL over its weights. The only task-specific component is the fitness function and a natural language description of the objective. The same framework, with essentially the same prompts, is applied to: 2D/3D finite point sets (exact star discrepancy fitness) and 32D Sobol direction numbers (rQMC MSE fitness on option pricing tasks).
>
>
> 3. However, about the mutation, does the LLM involve some kind of RL based training to generate better mutations.
>
> No. We do not train the LLM or apply RL. The LLM is used in a purely black-box way as a mutation operator within OpenEvolve: given the current population and fitness feedback, it is prompted to propose code edits, but its parameters are fixed. All “learning” happens via standard evolutionary selection at the program level. We have clarified this point more explicitly.

---

### Official Review · Reviewer_ihGh · 2025-11-01

**Soundness:** 2
**Presentation:** 3
**Contribution:** 1
**Rating:** 2
**Confidence:** 4

**Summary:**

This paper applies an LLM-guided evolutionary program synthesis framework (OpenEvolve) to two problems in Quasi-Monte Carlo (QMC) design:

**1**. constructing finite 2D/3D point sets with low star discrepancy


**2**. optimizing Sobol' direction numbers to reduce integration error for high-dimensional financial models.

The authors report finding new state-of-the-art 2D point sets for $N \ge 40$ and discovering new Sobol' parameters that outperform the standard Joe-Kuo parameters on a suite of financial option pricing tasks.

**Strengths:**

**1**. The paper successfully refined the Joe&Kuo's result, finding new Sobol's direction numbers that result in a statistically significant reduction in integration error (MSE) for a suite of 32-dimensional financial integration tasks;

**2**. The paper introduces the current LLM tools for solving a long-standing discrete optimization problem;

**3**. The paper is well-written, clearly organized and does a good job of introducing the complex technical details of QMC.

**Weaknesses:**

**Major Weaknesses:**


**1**.Minimal Contribution in Point Set Discovery:  The paper's "two-phase" strategy for point sets is a significant weakness. As shown in Figure 1, the LLM's "direct construction" in Phase 1 provides almost no improvement (0.0962 $\rightarrow$ 0.0924). The entire significant gain (0.0924 $\rightarrow$ 0.0744) comes from Phase 2, which is just the LLM generating code to call a standard classical optimizer (scipy.optimize.minimize). If this is the case, then it is not a novel discovery by the LLM; it's the automation of a standard workflow any human researcher would perform.

**2**. Lack of Methodological Novelty: The core evolutionary framework is a direct application of the pre-existing OpenEvolve. The paper does not propose any new methodological innovations to this framework. Its contribution is limited to applying this existing tool to the QMC domain.

**3**. Highly Incremental Sobol' Results: While statistically significant, the improvement in the Sobol' task is extremely small in absolute terms (e.g., an MSE of 4.52e-05 vs. 4.10e-05). Crucially, the search was initialized with the strong Joe & Kuo (2008) parameters. This frames the discovery not as a major breakthrough, but as a minor, local refinement of an existing solution.


**Minor Weakness:**

**1**. The authors acknowledge that they performed only one evolutionary run per problem. Since evolutionary algorithms are stochastic, this single run provides good new parameters but tells us nothing about the robustness or reliability of the search method.



If there's any misunderstanding, I would be more than happy to correct my opinions.

**Questions:**

I have two main questions, which are based on the weaknesses above:

**1.** Corresponding to Major Weakness 1, how could the authors justify that the improvement in the point-set problem comes from the LLM-guided approach, rather than just the application of scipy.optimize?

**2.** For the Sobol' discovery, how much credit belongs to the LLM's "intelligent search" versus the fact that it started with the Joe & Kuo parameters as its initialization? Have the authors tried running the search from a random initialization to see if it can discover good parameters from scratch?

---

> ### Author Response · Authors · 2025-11-26
>
> We thank the reviewer for their feedback, particularly regarding the source of improvement in point set discovery and the nature of the Sobol' optimization. These comments drove us to run additional validation/ablation analysis that helps clarify the paper's contribution.
>
> 1. Minimal Contribution in Point Set Discovery...it's the automation of a standard workflow any human researcher would perform.
>
> To address this, we implemented the exact baseline you suggested. In Appendix B.5, we compare our method against applying SLSQP directly to Sobol, Fibonacci, and Phase-I point sets without the LLM's evolutionary logic (Table 3). We did not include the points sets from Clement et al. as those are already locally optimal by design. The LLM-evolved programs achieved strictly lower star discrepancy than the pure SLSQP baselines in almost every case. We believe this demonstrates that the improvement is not solely due to the optimizer. The LLM discovers specific initialization strategies and restart heuristics that allow the classical optimizer to escape local minima better than standard applications of SLSQP.
>
> 2. Lack of Methodological Novelty ... limited to applying this existing tool to the QMC domain.
>
> We now acknowledge this in the Discussion:
>
> "Methodologically, our work does not introduce new evolutionary operators; our contribution is empirical and conceptual: we show that a generic LLM-guided program search, applied almost out-of-the-box, is capable of rediscovering and subtly improving long-studied QMC constructions, and we analyze when these gains go beyond what can be achieved with standard local optimization alone."
>
> More broadly, We view our contribution as primarily scientific and conceptual rather than methodological: we show that framing QMC design as program synthesis—letting an LLM search over the space of Python code that builds point sets and direction numbers—can recover known optima and push the frontier of best-known constructions in both 2D/3D finite sets and 32D digital sequences. We adapt OpenEvolve to a setting with expensive, mathematically defined fitness (exact star discrepancy and rQMC MSE with 10k scrambles) and demonstrate that the resulting programs are competitive with, and often better than, bespoke human-designed constructions.
>
> 3.  Highly Incremental Sobol' Results ... minor, local refinement of an existing solution.
>
> First, we agree that Joe–Kuo (2008) provides a very strong baseline, which is why we chose it as the starting point. Our intention is to show that even in this highly optimized regime, LLM-guided search can autonomously find further improvements.
>
> Table 3 in the main paper reports that for 32-dimensional Asian options, our evolved direction numbers reduce the MSE by roughly 9–10% relative to Joe–Kuo across several N, and this reduction is consistent across six scenarios (varying strike and volatility). This corresponds to ~10% fewer samples needed to achieve the same MSE, which is nontrivial in financial simulations. Beyond Joe–Kuo, Appendix C.1 compares our evolved parameters to Sobol nets obtained with LatNetBuilder, a specialized state-of-the-art tool for constructing digital nets. Our direction numbers achieve substantially lower rQMC MSE than these LatNetBuilder nets for the same sample sizes, especially at moderate N.
>
> Regarding initialization, Appendix C includes an ablation where we run the evolutionary search from randomly initialized direction numbers. This “LLM (random init.)” variant performs significantly worse than the Joe–Kuo-initialized run but matches the performance of naive Sobol' for large numbers of points, indicating that the LLM can in principle navigate the space from scratch, though warm-starting from a strong design is more efficient and yields the best results.
>
> 4. The authors acknowledge that they performed only one evolutionary run per problem ... tells us nothing about the robustness or reliability of the search method.
>
> We agree and have added analysis to demonstrate the robustness/reliability of the search method. In Appendix B.6 we run 16 independent seeds of the evolutionary search for N=100 and report the distribution of best-achieved discrepancies across seeds. We also show that a smaller LLM (Flash-Lite) and non-evolutionary prompting baselines perform consistently worse than the full evolutionary setup. In Appendix C.2, we have also included an ablation exploring the performance of discovered Sobol' direction numbers across 16 independent runs.

---

### Official Review · Reviewer_2Cbf · 2025-11-02

**Soundness:** 4
**Presentation:** 3
**Contribution:** 3
**Rating:** 8
**Confidence:** 4

**Summary:**

The paper applies OpenEvolve, a Large Language Model (LLM)-guided evolutionary program synthesis based on AlphaEvolve, to improve low-discrepancy sequences. Specifically, they apply this methodology to (1) construct finite 2D and 3D point sets with minimal star discrepancy (leading to small integration error); (2) optimising the direction numbers of Sobol’s sequence to reduce high-dimensional integration error
The LLM-guided evolutionary approach generates and mutates code that produces candidate point sets or sequence parameters, guided by a fitness function that measures, respectively, star discrepancy or integration error.
Specifically, in 2D, for a fixed number of points $N \leq 10$ the method recovered previously known optimal point sets and found new point sets with lower star discrepancy than previously discovered for $N > 30$ (0.0150 at $N=100$ vs the prior 0.0188).
For 3D sequences, the method matched all known optima up to N=7, and found new record-low discrepancies for N>8, for which optima are unknown.
Finally, for Sobol sequences in 32 dimensions, the evolved direction parameters reduced the mean squared error of randomised QMC integration on an Asian option pricing task compared to the standard Joe-Kuo parameters.

**Strengths:**

* The paper applies a powerful emerging method (LLM-guided evolutionary program synthesis) to a well-suited, long-standing problem in a creative manner.  This is very significant in two ways. Firstly, it is an interesting application of LLM-guided evolutionary methods, which are establishing themselves as an important tool in scientific discovery. Secondly, discovering new low-discrepancy point sets and sequence parameters is significant for the QMC and numerical methods community.

* The paper is clearly organised, provides an adequate amount of background, and contains extensive experimental results to back up its claims. The authors compare their results against a thorough set of baselines, including traditional sequences (Halton, Sobol, etc), simple lattice heuristics, and even more recent GNN-based MPMC methods.
While the Sobol sequence parameters were optimised with respect to the integration error of Asian option pricing, the authors evaluate their performance on multiple other exotic option types, showing that the solution generalises beyond this specific problem.

**Weaknesses:**

* Presentation:  line 267 has a missing reference. In addition, line 097 would benefit from a citation when referring to the “total variation of $f$ in the sense of Hardy and Krause”.

* The paper would benefit from more details on the LLM settings (which LLM is used?) and, perhaps, comparisons across multiple LLMs or LLM ensembles. I would expect that different LLMs would lead to different programs and solutions of different quality/optimality.

* A comparison or discussion justifying the choice of OpenEvolve over other LLM-guided evolutionary methods (such as ShinkaEvolve) would be interesting.

* Finally, one limitation compared to the previously established baseline Sobol parameters by Joe-Kuo is the solution’s potential specialisation for the options pricing problem.

**Questions:**

* Justify the choice of OpenEvolve instead of other LLM-guided evolutionary algorithms

*  Which LLM was used, what inference settings, and why?

---

> ### Author Response · Authors · 2025-11-26
>
> We thank the reviewer for the positive assessment and their insightful suggestions.
>
> 1. Presentation: line 267 has a missing reference. In addition, line 097 would benefit from a citation when referring to the “total variation of f in the sense of Hardy and Krause”.
>
> Thank you for catching these. The Koksma–Hlawka inequality and the “variation in the sense of Hardy and Krause” are now attributed to Koksma (1964), Hlawka (1961), and Dick & Pillichshammer (2010) in the introduction and background. The missing reference near line 267 was a formatting oversight that we have now fixed.
>
> 2. The paper would benefit from more details on the LLM settings (which LLM is used?) and, perhaps, comparisons across multiple LLMs or LLM ensembles. I would expect that different LLMs would lead to different programs and solutions of different quality/optimality.
>
> We have edited Section 4 (implementation details) and added Appendix A.3 to specify that we use the Gemini 2.0 Flash model as the mutation operator, with temperature 0.7 and top-p 0.95. We have added comparisons to the cheaper Gemini 2.0 Flash-Lite model in Appendix B.6 and Appendix C.2. The lighter model maintains improvement over classical baselines, and achieves performance close to the Flash model. Ensembles are an interesting next step to potentially generate more robust performance.
>
> 3. A comparison or discussion justifying the choice of OpenEvolve over other LLM-guided evolutionary methods (such as ShinkaEvolve) would be interesting.
>
> We have added text to the Discussion concerning this point:
>
> "Other more recent systems, such as ShinkaEvolve, extend the same underlying AlphaEvolve loop with additional engineering features tailored to long-horizon code benchmarks, such as richer parent sampling, novelty filtering, and advanced orchestration of program executions. In our setting, each candidate’s fitness is a cheap, deterministic quantity (star discrepancy or rQMC MSE) computed by running a short Python script, so the core evolutionary mechanism provided by OpenEvolve is sufficient to explore the search space effectively. We deliberately avoided any RL-style training or fine-tuning of the LLM: all improvements come from search in program space driven by an off-the-shelf model and the numerical fitness signal."
>
> In short, we believe ShinkaEvolve could be applied and achieve similar success, but find OpenEvolve sufficient for our tasks.
>
> 4. Finally, one limitation compared to the previously established baseline Sobol parameters by Joe-Kuo is the solution’s potential specialisation for the options pricing problem.
>
> We agree that over-specialization is a concern and so we evaluate the evolved Sobol' parameters on a suite of six Asian option scenarios (varying moneyness and volatility) and on several other exotic payoffs (lookback, basket, and Bermudan options). In all smooth-payoff cases, the new parameters match or improve upon Joe–Kuo. In general, we find the optimal choice of Sobol' parameters to be highly dependent on the problem at hand; there is some tradeoff between robustness to different problems and performance on any one specific problem. In practice, we believe this framework could be applied to generate custom Sobol' direction numbers suited to any problem of interest, such as global sensitivity analysis of a specific simulation.

---

> > ### Comment · Reviewer_2Cbf · 2025-11-27
> >
> > thank you for clarifying my points; I will keep my positive score

---

### Author Response · Authors · 2025-11-26

We thank the reviewers for their constructive and rigorous feedback. Based on your comments, we have significantly revised the paper to include new baselines and ablation studies that isolate the specific contributions of the LLM and the evolutionary search.

Key Updates in the Revision:

Isolation of LLM Contribution vs. Local Optimization: We added a baseline in Appendix B.5 comparing our method against classical optimization (scipy.optimize / SLSQP) applied to Sobol, Fibonacci, and Phase-I seeds. Our results (Table 3) show that the LLM-guided approach finds lower discrepancy configurations than local optimization alone, demonstrating that the LLM contributes non-trivial initialization and restart strategies.

Comparison to State-of-the-Art QMC Tools: We added a comparison against LatNetBuilder (Appendix C.1, Table 6), a software for QMC design. Our evolved Sobol’ parameters achieve lower rQMC MSE for sample sizes N≤1024 and remain competitive for larger N.

Ablation of Evolutionary Search vs. Prompting: We added Appendix B.6 (Table 4), comparing the full evolutionary loop against single-turn and multi-turn prompting strategies. The results confirm that population-based evolution yields consistently better and more robust solutions than repeated prompting.

Role of Initialization: We investigated learning from scratch vs. warm-starting (Appendix C.2). We confirm that initializing with Joe & Kuo parameters is crucial for efficiency, framing our method as a refinement of expert priors rather than ab initio discovery for high-dimensional sequences.

Computational Complexity: We added Appendix A.4, providing a detailed breakdown of token costs, execution time, and fitness evaluation budgets.

---

### Meta-Review · Area_Chair_LaeF · 2026-01-07

**Summary:**

This paper features an interesting contribution in which the classical QMC design is now framed as program synthesis. In this view, an LLM search over the space of Python code that builds point sets and direction numbers to recover known optima and push the frontier of best-known constructions in both 2D/3D finite sets and 32D digital sequences. This highlights an emergent capability of LLM as a reasoning/searching tool over a concept space (i.e., Python program) to boost scientific discovery. The authors adapt OpenEvolve to a setting with expensive, mathematically defined fitness (exact star discrepancy and rQMC MSE with 10k scrambles) and demonstrate that the resulting programs are competitive with, and often better than, bespoke human-designed constructions.

The opinions of the reviewers on this paper are somewhat mixed. Two reviewers rate this work highly (8) and provide favorable feedback, which highlights well the conceptual elegance of this work.

On the other hand, reviewer (ihGh) raised concerns regarding the technical novelty in the sense that this work does not create new LLM technologies. The reviewer also raised a concern regarding the marginal improvement over an already strong baseline of point set discovery; as well as the marginal improvement of the construction phase.

Reviewer W9ms acknowledged that casting QMC design as program synthesis is conceptually elegant and connects symbolic LLM reasoning with continuous numerical optimization, allowing for a well-calibrated combination of constructive heuristics with iterative optimization. However, the reviewer was cautious that the problem setting is somewhat narrow and there might be issue with scalability to solving higher dimensional problem with this LLM-guided representation.

Reviewer jdvu argued that this work is limited in scientific evaluation. For example, comparing with simpler methods for exploring the space of low-discrepancy sequences, reporting the improvement in terms of fitness during the run of the evolutionary algorithm, comparing with simple (single- or multi-turn) LLM prompting without population-based evolution.

--

The authors have provided detailed rebuttal which, in my opinion, address well the concerns from the three critical reviewers.
Reviewer jdvu confirmed that the added experiments have address most of the concerns on evaluation (though the reviewer remains skeptical that the improvement is somewhat marginal).

The added experiments also extensively address the concerns of reviewers W9ms and ihGh. Specifically, the authors have expanded the experiments on other problems (addressing the criticism on narrow problem setting) as well as constructed ablation studies that isolate and demonstrate the impact of the construction phase consistently across multiple problem setting (addressing reviewer ihGh's concerns on the marginal impact of the construction phase).

The new experiments have also tested the method on problem settings with relatively higher dimension, partly addressing W9ms's concern on scalability.

--

Overall, the remaining concerns are: (1) this work does not create new LLM technologies (reviewer ihGh); (2) limited scientific improvement (reviewer jdvu); and (3) scalability to tasks with much higher dimension. In my opinion, (1) and (2) are orthogonal to the main message here which is that classical QMC design is now framed as program synthesis via an emergent capability of LLM as a reasoning/searching tool over a concept space to boost scientific discovery. In other words, these criticisms arise from a narrow LLM and/or pure scientific views of this work which miss the above big picture. On the scalability, I'd take this as an area of improvement for the current work rather than a critical flaw.

--

With the above rationale, I am happy to recommend acceptance for this paper.

**Reviewer Concerns:**

Please refer to my assessment in the summary. Overall, the rebuttal has addressed most technical concerns and the remaining ones are orthogonal to the main contribution so I'd not take them as show stoppers.

**Reviewer Scores:**

Following the rebuttal, the two reviewers who gave a rating of 8 retain their scores as expected. Reviewer jdvu raised his score from 2 to 4, acknowledging that the main concerns were addressed but he remains concerned with the limited improvement from a strict scientific view of the solved problems. This is true but as I mentioned, it is a narrow view of the paper which fails to acknowledge the cross-domain impact of using LLM to reframe long-standing challenges in QMC as program synthesis. Likewise, the remaining concerns of reviewer ihGh regarding having no creation of new LLM technologies fall into the pure LLM view which also misses the above cross-domain impact. As these criticisms are heavily dependent on the reviewers' perspective, I am not sure they would change their minds (and I'd respect that) but I expect that the two positive reviewers would champion for this paper and I'd concur with them. Finally, given the extensive rebuttal that largely expand on multiple tasks, I believe reviewer W9ms will increase the score as the concern on narrow problem setting has been addressed. The overall stance is thus positive in my opinion.

---

### Decision · Program_Chairs · 2026-01-26

Accept (Poster)